# Water isotopic characterisation of the cloud-circulation coupling in the North Atlantic trades. Part 2: The imprint of the atmospheric circulation at different scales

Leonie Villiger[1,2] and Franziska Aemisegger[1]

[1]Institute for Atmospheric and Climate Science, ETH Zurich, Zurich, Switzerland
[2]Institute for Environmental Decisions, ETH Zurich, Zurich, Switzerland

**Correspondence:** Leonie Villiger (leonie.villiger@env.ethz.ch)

**Abstract.** Water vapour isotopes reflect the history of moist atmospheric processes encountered by the vapour since evaporating from the ocean, offering potential insights into the controls of shallow trade-wind cumuli. Given that these clouds, particularly their amount at the cloud base level, play an important role in the global radiative budget, improving our understanding of the hydrological cycle associated with them is crucial. This study examines the variability of water vapour isotopes at cloud base in the winter trades near Barbados and explores its connection to the atmospheric circulations ultimately governing cloud fraction. The analyses are based on nested COSMO$_{iso}$ simulations with explicit convection during the EUREC[4]A field campaign. It is shown that the contrasting isotope and humidity characteristics in clear-sky versus cloudy environment at cloud base emerge due to vertical transport on time scales of 4 to 14 hours associated with local, convective circulations. In addition, the cloud base isotopes are sensitive to variations in the large-scale circulation on time scales of 4 to 6 days, which shows on average a Hadley-type subsidence but occasionally much stronger descent related to extratropical dry intrusions. This investigation, based on high-resolution isotope-enabled simulations in combination with trajectory analyses, reveals how dynamical processes at different timescales act in concert to produce the observed humidity variations at the base of trade-wind cumuli.

## 1 Introduction

The response of shallow trade-wind clouds to climate change is uncertain and known to contribute substantially to the spread of climate projections (e.g., Bony et al., 2015; Zelinka et al., 2017). Especially, the cloud fraction at cloud base has been identified as a key variable influencing the spread of the modelled feedback of these clouds to climate change (Bony et al., 2017). To shed light on the mechanisms controlling cloud base cloud fraction in the trades, the field campaign EUREC[4]A (Stevens et al., 2021) was conducted in early 2020 near Barbados. The collected observations highlight the role of shallow mesoscale circulations (George et al., 2023) in driving the variability of vertical velocities at cloud base, which is an important control of cloud fraction at this level (Vogel et al., 2022). It remains to be investigated how these circulations, which have recently gained attention from the scientific community, shape the environment, particularly the distribution of humidity (Albright et al., 2022) and eventually cloud fraction.

Here, we are interested in using the abundance of heavy stable water vapour isotopologues as tracers of cloud microphysical processes, as well as turbulent and convective mixing. Heavy stable water vapour isotopologues (hereafter isotopes) are water molecules containing a heavy hydrogen ($^1H^2H^{16}O$ or HDO) or oxygen atom ($^1H_2^{18}O$). Compared to their light counterpart ($^1H_2^{16}O$), they have lower saturation vapour pressures and lower diffusion velocities. This implies that the heavy water molecules preferably stay in the condensed phase, where they establish stronger intermolecular bonds compared to their lighter counterpart (equilibrium fractionation). Furthermore, the near-surface humidity gradient leads to a differentiation in the relative concentration of the two heavy isotopes ($^1H^2H^{16}O$ and $^1H_2^{18}O$) due to their differences in diffusivity (non-equilibrium fractionation). This results in a change in the relative abundance of heavy-to-light isotopes during phase transitions. The isotopic composition of a water sample is typically assessed with the $\delta$-value for $^2H$ and $^{18}O$, respectively:

$$\delta^2H\ [\text{‰}] = \left( \frac{^2R_{\text{sample}}}{^2R_{\text{VSMOW}}} - 1 \right) \cdot 1000 \tag{1}$$

$$\delta^{18}O\ [\text{‰}] = \left( \frac{^{18}R_{\text{sample}}}{^{18}R_{\text{VSMOW}}} - 1 \right) \cdot 1000 \tag{2}$$

The $R$ in the equations above stands for the atomic ratio of the concentration of the heavy to the light isotope, namely $^2R = \frac{[^2H]}{[^1H]}$ and $^{18}R = \frac{[^{18}O]}{[^{16}O]}$, in the water sample and the internationally accepted Vienna Mean Ocean Standard Water 2 (VSMOW; International Atomic Energy Agency, 2017). The relative variations of $\delta^2H$ and $\delta^{18}O$ due to non-equilibrium fractionation is assessed with the deuterium excess:

$$\text{d-excess} = \delta^2H - 8 \times \delta^{18}O \tag{3}$$

which is a measure of the thermodynamic disequilibrium of the environment during phase transitions (e.g., Pfahl and Wernli, 2008). Due to these mechanisms, the abundance of heavy isotopes reflects the integral of all phase changes and mixing processes that occur along the flow. While the first-order isotope variable $\delta^2H$ is sensitive to microphysical processes and mixing (Gat, 1996; Galewsky et al., 2016), the second-order isotope variable d-excess is sensitive to the thermodynamic conditions at the moisture source (Pfahl and Wernli, 2008; Aemisegger et al., 2021).

Part 1 of this paper showed that the horizontal variability of humidity and $\delta^2H$ of vapour at the cloud base emerges from the circulation involving ascent within convective clouds and subsidence outside of clouds, which we will henceforth refer to as the cloud-relative circulation (Fig. 1a). Cloudy cloud base patches (representing the ascending branch) are moister and more enriched than the background conditions. In contrast, clear-sky, dry-warm cloud base patches (hereafter dry-warm patches; representing the descending branch) are drier and more depleted than the background conditions. This suggests, against the above-formulated expectations, that the isotopic characteristics of the two cloud base features mainly reflect vertical transport and are not primarily controlled by local microphysical or turbulent mixing processes, an aspect that we will investigate in more detail in part 2 of this study.

In this paper, we establish a link between isotopes in the trade-wind region and the characteristics of atmospheric circulations. For this, we use three nested convection resolving COSMO$_{\text{iso}}$ simulations with different resolutions and air parcel backward

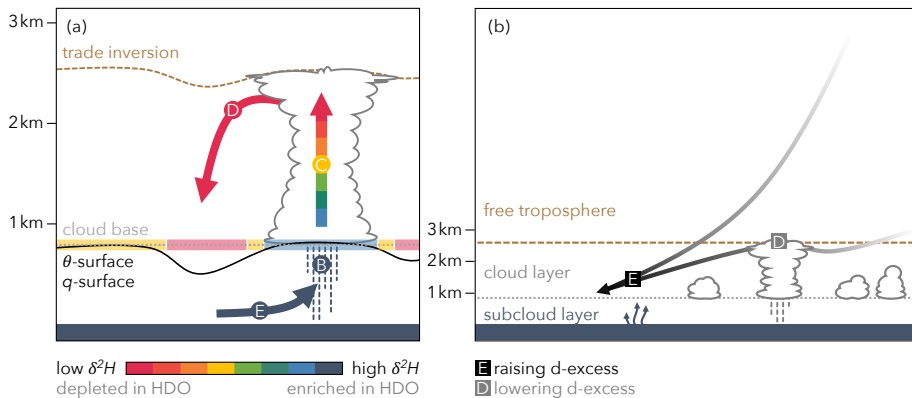

**Figure 1.** Idealised schematic of the processes affecting **(a)** the $\delta^2$H in vapour within the circulation associated with clouds, and **(b)** the d-excess in vapour during large-scale transport. Shown are three atmospheric layers, the subcloud layer, the cloud layer, and the free tropo­sphere. They are separated by the cloud base level (dotted grey) and the trade inversion (dashed brown). It is assumed that the two boundaries have an uneven topography in reality. However, in the simulations, cloud base is identified at a constant height. Therefore, the *cloud* (trans­parent blue in a), the *clear* (transparent yellow in a) and the *dry-warm* (transparent red in a) cloud base environments are defined at the flat cloud base (see Sect. 2.3 for the detailed definition of the three environments). **(a)** Within the cloud-relative circulation, air parcels (E) take up freshly evaporated and, therefore, isotopically heavy vapour (high $\delta^2$H; enriched in $^1$H$^2$H$^{16}$O) from near the ocean surface; (B) may encounter below-cloud processes such as partial or full evaporation of hydrometeors and equilibration between vapour and liquid droplets, which can have a depleting or an enriching effect on the vapour (depending on the saturation level of the subcloud layer and the forma­tion altitude of the rain, Aemisegger et al., 2015; Graf et al., 2019); (C) will continuously lose heavy isotopes (lowering $\delta^2$H; depleted in $^1$H$^2$H$^{16}$O) as soon as they reach the lifting condensation level and cloud and rain droplets are formed. Note that the temperature effect makes the fractionation stronger with increasing altitude. At any height, the now isotopically light air parcels may be detrained from the cloud into the surrounding clear-sky environment. Here, (D) the air parcels' vapour can get further depleted by mixing with vapour from above the trade inversion. **(b)** Within the large-scale circulation, air parcels can (E) take up moisture that is freshly evaporated from the ocean surface under non-equilibrium conditions and therefore has a relatively high d-excess (Pfahl and Wernli, 2008); (D) get moistened through the detrainment of cloudy air from precipitating clouds and have a comparatively low d-excess (Noone, 2012; Thurnherr and Aemisegger, 2022).

55 trajectories from the cloud base environment. This data set, and the applied methods, are described in detail in Sect. 2. First, we look at processes on the subdaily timescale (Sect. 3). We investigate how convection drives the variability of humidity and isotopes in different cloud base environments (cloudy vs. dry-warm). Since convection in the trades has a clear diel cycle (Vial et al., 2019; Vogel et al., 2020; Vial et al., 2021), we use the diel cycle as a framework to answer the question: *Does the diel cycle of humidity and $\delta^2$H in different cloud base environments reflect the growth and decay of convection?* Second, we test

60 the hypothesis that the large-scale circulation transporting air into the trade-wind region leaves a distinct isotope signal in the cloud base vapour (Sect. 4). We thus address the question: *Which of the three variables, specific humidity (q), $\delta^2$H, and d-excess is most strongly influenced by the large-scale circulation?* In the conclusion, we combine the findings from the two research questions (Sect. 5).

## 2 Data and methods

### 2.1 Datasets

The data from three convection-resolving COSMO$_{iso}$ simulations, described and evaluated in Villiger et al. (2023), are used. Convection-resolving means that the convection schemes of the model (parameterising deep and shallow convection; see Tiedtke, 1989; Theunert and Seifert, 2006) were disabled. We have used the COSMO$_{iso}$ model in this setup in many previous studies (Dahinden et al., 2021; Diekmann et al., 2021; Thurnherr et al., 2021; de Vries et al., 2022), in which the comparison with isotope observations in various regions of the world have shown a good performance of the explicit convection setup. Furthermore, previous analyses have shown that COSMO simulations at a range of resolutions (grid spacing $\leq 25\,km$) do not necessarily provide more realistic results in terms of radiation and precipitation patterns if shallow convection is parameterised (Vergara-Temprado et al., 2020).

For the three simulations used here, a $20\,s$ model timestep was applied and hourly output was generated. The simulations differ in terms of domain as well as horizontal (10, 5, 1 km) and vertical (40, 60, 60 levels) grid spacing. These differences are summarised below (for more details see Villiger et al., 2023, in particular their Fig. 2):

- COSMO$_{iso,\,10km}$ has a horizontal resolution of $0.1°$, 40 vertical levels, and covers most of the North Atlantic. Horizontal winds above $850\,hPa$ in COSMO$_{iso,10km}$ were nudged towards a simulation performed with the global model ECHAM6-wiso (Cauquoin et al., 2019; Cauquoin and Werner, 2021), which also served as a source for initial, lateral, and top boundary conditions at 6-hourly intervals. A Rayleigh damping to the top boundary condition was used in the upper levels of the model domain. The ECHAM6-wiso simulation, itself, was nudged towards ERA5 (Hersbach et al., 2020) to reproduce the large-scale meteorological conditions of the simulated period. COSMO$_{iso,10km}$ covers the period from 6 January to 13 February 2020 of which the first 10 days are treated as spin up and are not included in the analysis.

- COSMO$_{iso,5km}$ has a horizontal resolution of $0.05°$, 60 vertical levels, and covers a subset of the western North Atlantic, including the northern part of the South American continent. Initial and lateral boundary conditions originate from COSMO$_{iso,10km}$ at hourly timesteps. The spectral nudging technique was identical to the first COSMO$_{iso}$ simulation but nudged towards COSMO$_{iso,10km}$ horizontal winds above $850\,hPa$ instead of ECHAM6-wiso. COSMO$_{iso,5km}$ covers the period from 20 January to 13 February 2020. All simulated days are included in the analysis.

- COSMO$_{iso,1km}$ has a horizontal resolution of $0.01°$, 60 vertical levels, and covers the focus area of the EUREC[4]A campaign's field activity. Initial and lateral boundary conditions originate from COSMO$_{iso,5km}$ at hourly timesteps and the spectral nudging was directed towards COSMO$_{iso,5km}$ wind data. COSMO$_{iso,1km}$ covers the period from 20 January to 13 February 2020, and all simulated days are taken into account for the analysis.

COSMO$_{iso,1km}$ and COSMO$_{iso,5km}$ are used to characterise the cloud-relative circulation (Sect. 3), and COSMO$_{iso,10km}$ to assess the large-scale circulation (Sect. 4). For this, air parcel backward trajectories are calculated with data from COSMO$_{iso,5km}$ and COSMO$_{iso,10km}$ using the Lagrangian analysis tool LAGRANTO (Wernli and Davies, 1997; Sprenger and Wernli, 2015,

see detailed description of trajectory starting points in Sect. 2.4 and Sect. 2.5). LAGRANTO trajectories are based on the three-dimensional hourly wind fields of the respective dataset, which do not resolve sub-grid scale boundary layer processes. To alleviate this limitation, we compute a large set of trajectories. Note also that the COSMO$_{iso,10km}$ trajectories spend comparably little of their lifetime near the boundary layer, and the COSMO$_{iso,5km}$ trajectories are explicitly started from features that are
dominated by subsidence and thus well characterised by the grid-scale winds.

## 2.2 Definition of cloud base

Cloud base is identified using the same procedure as in Villiger et al. (2023), which includes the following steps:

1. Vertical profiles at every grid point in the domain 54.5-61° W and 11-16° N are checked for cloud water content exceeding $10\,\mathrm{mg\,kg^{-1}}$ (threshold for the detection of clouds used in Vial et al., 2019) at any model level below $1.3\,\mathrm{km}$. The
lowest model level meeting this criterion is taken as the cloud base of the respective vertical profile. If a given profile does not contain any clouds (i.e., does not meet the criteria), it is ignored in the subsequent step.

2. In order to extract cloud base conditions, we determine one representative cloud base model level for the domain 54.5-61° W and 11-16° N by calculating the median over the cloud base model levels identified for cloudy profiles in the previous step.

3. Steps 1 and 2 are repeated for every hourly timestep of the simulated time period. The resulting time series of cloud base model levels can then be used to extract cloud base variables from the COSMO$_{iso}$ simulations.

The hourly cloud base levels alternate between 783 and $970\,\mathrm{m}$ for COSMO$_{iso,10km}$ and change between three levels, i.e. 761, 914, and $1082\,\mathrm{m}$, for COSMO$_{iso,5km}$ and COSMO$_{iso,1km}$.

## 2.3 Definition of cloud base features

The COSMO$_{iso,1km}$ and COSMO$_{iso,5km}$ grid points at cloud base in the domain 54.5-61° W and 11-16° N are assigned to three categories representing features from the circulation associated with clouds, clear-sky regions with dry-warm anomalies, and clear-sky regions without dry-warm anomalies (see details in Villiger et al., 2023, note that here we do not differentiate between precipitating and non-precipitating clouds). The following definitions are applied to assign the data points to the three categories *cloud*, *dry-warm*, and *clear*:

– *Cloud* grid points are identified based on liquid cloud water content exceeding $10\,\mathrm{mg\,kg^{-1}}$.

– *Dry-warm* grid points are identified by a positive anomaly in potential temperature ($\theta$) and a negative anomaly in $q$. The anomalies are defined grid-point-wise by removing the daily cycle. For each grid point, the hour-of-the-day mean and standard deviation are calculated over the whole simulated period (20 January to 13 February 2020). *Dry-warm* grid points are then selected using the following criteria: $q_{i,t} < \overline{q_{i,h(t)}} - \sigma(q_{i,h(t)})$ and $\theta_{i,t} < \overline{\theta_{i,h(t)}} + \sigma(\theta_{i,h(t)})$ ($i$ denoting
the grid points inside the evaluation domain, $t$ the hourly timesteps of the simulated period, $h(t)$ the hour of the day

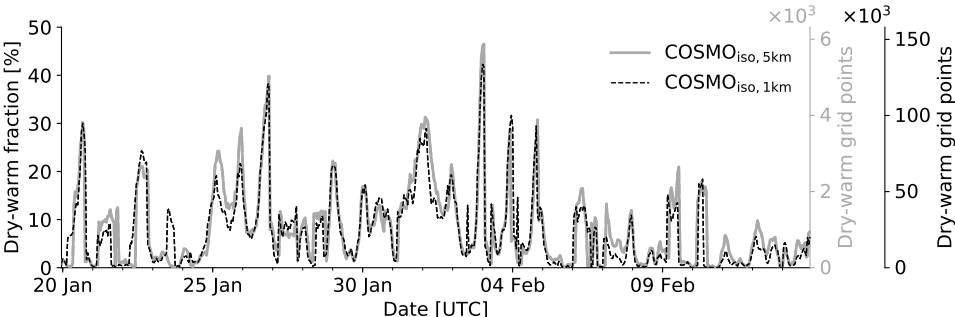

**Figure 2.** Hourly time series of fraction (left axis) and number (right axis) of cloud base grid points in the domain 54.5-61° W and 11-16° N categorised as *dry-warm* in COSMO$_{iso,5km}$ (gray continuous) and COSMO$_{iso,1km}$ (black dashed). In case of COSMO$_{iso,5km}$, these cloud base grid points serve as starting points of the *dry-warm* backward trajectories (see text for details). The total number of cloud base grid points in the considered domain are 12'632 for COSMO$_{iso,5km}$, and 316'028 for COSMO$_{iso,1km}$.

corresponding to the timestep $t$, and $\sigma$ the standard deviation of the considered variable). We checked that these criteria exclusively select clear-sky grid points (i.e., liquid cloud water content below $10\,\mathrm{mg\,kg^{-1}}$) without using an additional criterion for the liquid cloud water content. An overview of the number of identified *dry-warm* grid points per timestep is given in Fig. 2 and an exemplary timestep illustrating their spatial distribution in Fig. 3.

– The remaining clear-sky grid points (i.e., liquid cloud water content below $10\,\mathrm{mg\,kg^{-1}}$, but no positive anomaly in $\theta$ combined with a negative anomaly in $q$) are categorised as *clear*.

The reasoning behind the separation of *dry-warm* and *clear* grid points at cloud base into different categories stems from part 1 of this study. We assume that the characteristics of the *dry-warm* category result from coherent mesoscale subsidence and, therefore, can give insight into the downward branch of the cloud-relative circulation (sketched in Fig. 1). A *dry-warm* anomaly

is expected at the cloud base level of the downward branch because (1) subsidence causes adiabatic warming (generating a warm anomaly at cloud base), and (2) a coherent mesoscale *dry-warm* anomaly ensures a certain distance from clouds and through this minimises the influence of mixing with moist air from surrounding clouds thereby avoiding major impacts of evaporating cloud and rain droplets. Crucially, the absence of mixing and phase changes along the subsidence path leads to a conservation of the isotope signal in the vapour from the point at which it was detrained from the cloud down to cloud base.

As shown in the exemplary timestep in Fig. 3, our definition of *dry-warm* indeed applies to grid points that are well away from clouds and therefore are optimal to analyse the processes associated with subsidence alone, which we expect to be resolved in the simulations used here.

For the *clear* category, we assume that several processes (subsidence, turbulent mixing, evaporation of cloud and rain droplets) impact its characteristics. Some of these, e.g., turbulent mixing, occur on shorter temporal and spatial scales than

resolved by our backward trajectories based on hourly simulation output. Whether the two cloud base environments, *clear* and

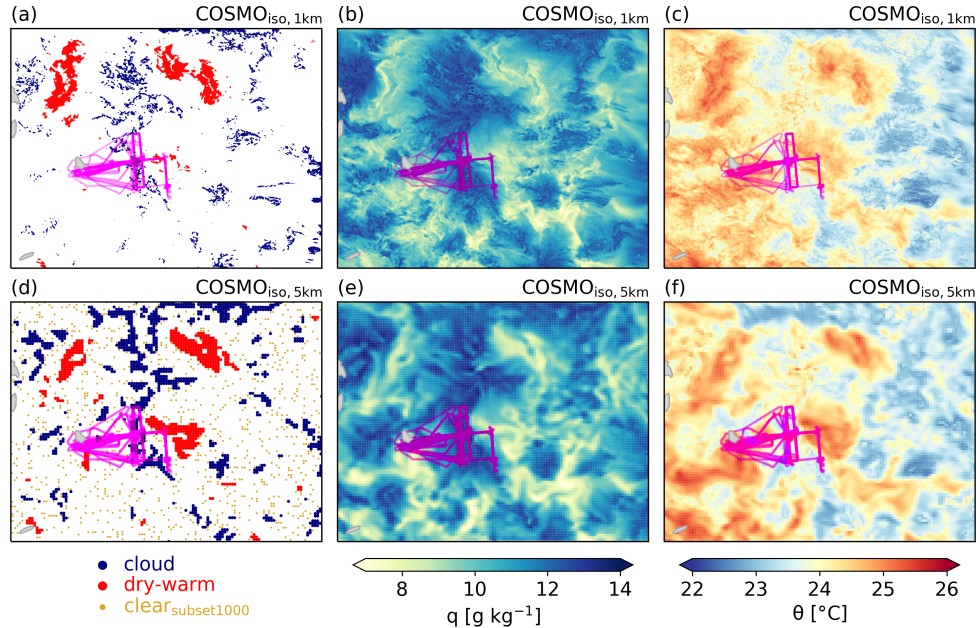

**Figure 3.** Spatial distribution of **(a,d)** cloud base grid points identified as *cloud* (blue) *dry-warm* (red) or *clear* (yellow; only a subset of 1000 randomly selected data points), **(b,e)** specific humidity ($q$), and **(c,f)** potential temperature ($\theta$) at cloud base at 15 UTC on 2 February 2020. Shown is the data from **(a-c)** COSMO$_{iso,1km}$ and **(d-f)** COSMO$_{iso,5km}$ in the domain 54.5-61° W and 11-16° N. The fraction of grid points in the domain identified as *dry-warm* is **(a)** 3.3 % and **(d)** 4.3 %. **(d)** COSMO$_{iso,5km}$ backward trajectories (see text for details) are started from the red areas (representing all *dry-warm* cloud base grid points) and the yellow dots (representing 1000 randomly selected *clear* grid points at cloud base). For scale, the flight track of the aircraft ATR (Bony et al., 2022) during EUREC[4]A is shown in pink.

*dry-warm*, truly emerge due to different processes is assessed statistically using backward trajectories as described in the next section.

    Although we have a special interest in the *dry-warm* category, it is important to remember that *clear* grid points at cloud base cover a much larger area. Namely, about 81 % of the cloud base grid points in COSMO$_{iso,1km}$ are categorised as *clear* and only about 8 % as *dry-warm*, considered over the whole simulated period. This means, for instance, that for the mass balance at cloud base, the *clear* category plays a more important role than the *dry-warm* one (knowing that at the cloud base level, local downward winds of similar magnitude prevail in both; see Villiger et al., 2023, their Fig. 15d).

## 2.4 Lagrangian characterisation of the cloud-relative circulation

To investigate the formation mechanism of the *dry-warm* patches at cloud base, we calculate 24 h-backward trajectories. We start them at hourly timesteps between 22 January and 13 February 2020 from all cloud base grid points identified as *dry-warm* in the domain 54.5-61° W and 11-16° N in the COSMO$_{iso,5km}$ simulation. Note that the number of *dry-warm* cloud base grid points varies from timestep to timestep (Fig. 2). Summed over all timesteps, this results in a total of 568'124 *dry-warm*

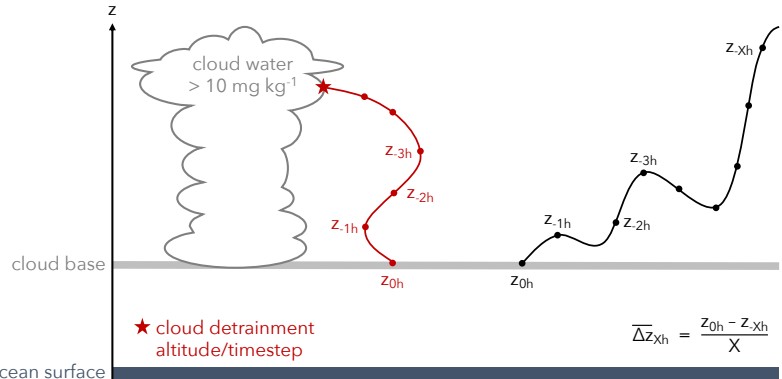

**Figure 4.** Schematic of two individual air parcels' backward trajectory started from cloud base. The **red trajectory** showcases the procedure to identify the cloud detrainment altitude and timestep of the air parcel (red star). This procedure is applied to the $COSMO_{iso,5km}$ backward trajectories (Sect. 2.4). The **black trajectory** showcases the procedure to obtain the mean vertical displacement over a certain time period ($\overline{\Delta z}_{Xh}$), which is calculated for the $COSMO_{iso,10km}$ backward trajectories (Sect. 2.5). Here, $z_{0h}$ indicates the altitude from which the backward trajectory is started, and $z_{-1h}$, $z_{-2h}$, $z_{-3h}$, ... , $z_{-Xh}$ the altitude of the air parcel 1 h, 2 h, 3 h, ... , X h before arrival at cloud base, respectively. The altitude difference between $z_{0h}$ and $z_{-Xh}$ divided by the hours between the two considered timesteps yields the mean altitude change over X h, i.e., $\overline{\Delta z}_{Xh}$. Note that if time periods longer than 24 h are considered, the notation is adapted to days instead of hours (i.e., $\overline{\Delta z}_{Xd}$).

trajectories. Similarly, we start 24 h-backward trajectories every hourly timestep between 22 January and 13 February 2020 from 1000 randomly selected cloud base grid points identified as *clear* in the $COSMO_{iso,5km}$ simulation. We fix the number of *clear* trajectories for computational reasons, since about 9'000 grid points are identified as *clear* every hourly timestep (not shown). Summed over all timesteps, this results in a total of 539'460 *clear* trajectories. The starting points of the *dry-warm* (red areas) and *clear* (yellow dots) backward trajectories are shown for an exemplary timestep in Fig. 3d.

To learn about the overturning time and vertical depth of the cloud-relative circulation, we determine the altitude and timestep, at which each air parcel was last located inside a cloud before arriving at cloud base (i.e., we perform a last point of saturation analysis, cf. Galewsky et al., 2016). For this, we check the cloud liquid water content along each trajectory and identify the last timestep before arrival, when the cloud liquid water content exceeded $10\,\mathrm{mg\,kg^{-1}}$ (sketch in Fig. 4). We refer to the air parcels' altitude at this timestep as cloud detrainment altitude (Fig. 5c and Fig. 10a) and interpret it as the upper turning point of the circulation (where the air parcel is detrained from the cloudy updraft and starts to subside towards the surface). Note that only 58 % of the air parcels arriving in a *dry-warm* cloud base environment encounter a cloud during the previous 24 h. For air parcels arriving at *clear* grid points at cloud base, this value amounts to 79 %.

We limit the trajectories to 24 h because we want to isolate the coupling between the upward and downward branch of the circulation. Since the upward branch (i.e., convection) is known to have a clear diel cycle (Vial et al., 2019, 2021; Vogel et al., 2020, 2022), it is reasonable to look at the downward branch over the same time period. We use $COSMO_{iso,5km}$ for these trajectories because $COSMO_{iso,1km}$ has too small a domain to trace air parcels over several hours. We use the variables cloud detrainment altitude and timestep in Sect. 3 and 5.

## 2.5 Lagrangian characterisation of the large-scale circulation

We calculate backward trajectories based on $COSMO_{iso,10km}$ data to evaluate the role of the large-scale circulation for the isotopic composition of vapour around cloud base in Sect. 4. A total of 138 trajectories, reaching $6\,d$ backwards in time, are started every hour from 22 January to 13 February 2020, which is sufficient to capture the influence of a large-scale signal. The starting points are distributed over three vertical levels and horizontally spaced with a distance of $100\,km$ in the domain $54.5$-$61°$ W and $11$-$16°$ N. The three vertical levels at 940, 920, and $900\,hPa$ are chosen such that they bracket cloud base to take into account some variability in the cloud base level in the coarse resolution dataset. Hereafter, the air parcels' arrival level is referred to as cloud base. Furthermore, we do not distinguish between different cloud base mesoscale features (as in Sect. 2.3 and Sect. 2.4), because we expect the large-scale circulation to modulate isotope signals at the large-scale.

To distinguish between different large-scale flow patterns and to assess the coupling between large-scale flow and cloud base isotopes, we calculate the mean $1$-$h$ vertical displacement based on the change of altitude ($\overline{\Delta z}_{Xd}$; sketch in Fig. 4) over different time windows ($X = 1, ..., 6\,d$) for each trajectory. This variable is used in Sect. 4 and 5.

The sources of the moisture arriving in the EUREC$^4$A domain and their associated conditions are characterised based on the algorithm developed by Sodemann et al. (2008) and using the $COSMO_{iso,10km}$ trajectories. In short, a trajectory-based water mass balance is computed and temporal changes in $q$ along the backward trajectories are identified as uptakes if positive and rain events if negative. The weight of each uptake is determined according to its mass contribution to the final $q$ at arrival by proportionally discounting the influence of uptakes happening before rain events underway. Moisture source conditions, in particular the relative humidity with respect to sea surface temperature ($RH_{SST}$), which is known to control the d-excess signal, were calculated as mass weighted means using the weights of the individual uptakes (for more details, see Aemisegger et al., 2014).

## 3 Cloud base isotopes and the cloud-relative circulation

This section discusses how the cloud-relative circulation (Fig. 5) drives the diel cycle of isotopes in different cloud base environments (Fig. 6). For this, we analyse the temporal evolution of the isotope signals at the cloud base data points from $COSMO_{iso,1km}$ categorised as *cloud*, *clear*, and *dry-warm* (Sect. 2.3). In addition, we use $COSMO_{iso,5km}$ trajectories to derive an estimate of the rooting altitude of the subsiding branch of the cloud-relative circulation (cloud detrainment altitude, see Sect. 2.4). When combining these two datasets, it is important to know that the vertical velocities within *cloud* and *dry-warm* cloud base environments are stronger in $COSMO_{iso,1km}$ than in $COSMO_{iso,5km}$. However, the diel cycles of vertical velocities in the two datasets are the same (not shown). We argue that we can use the variables characterising the cloud-relative circulation derived from the $COSMO_{iso,5km}$ trajectories, as our primary focus lies in discerning their variation (and not their absolute values) throughout the day and in understanding how this variation correlates with changes in cloud base isotopes.

Before we move on to exploring the potential coupling between the cloud-relative circulation and cloud base isotope signals, we have to briefly address the evaluation of the diel cycles with observations. The modelled diel cycles cannot be evaluated directly with observations due to the lack of available observations at cloud base in different environments (*cloud*, *clear*, *dry-*

*warm*) over the entire day. Instead, an evaluation of the diel cycle from observations at a close-by land site along the east coast of Barbados (BCO site, see Bailey et al., 2023) has been done, which shows a good agreement with the model (Appendix A). The diel cycles of $q$ and $\delta^2H$ in clouds at cloud base are in phase with and of the same amplitude as the respective variable at the coastal site. A slight delay in phasing of the diel cycles in the *clear* and *dry-warm* patches at cloud base compared to the near-surface coastal point is due on the one hand to the circulation including the condensational depletion in clouds (see also discussions below), and, on the other hand, to the stronger direct influence of surface evaporation as well as below cloud interaction with falling rain at the BCO. From this short comparison with the observational BCO data, we conclude that the physical ingredients shaping the diel cycle are adequately represented in the model.

We start the discussion of the diel cycles by considering the variations of cloud fraction and precipitation over a typical day. The cloud fraction at cloud base is maximal at night between 22 and 02 local time (LT; 02-06 UTC), followed by a continuous decrease until reaching a minimum shortly after noon between 13 and 15 LT (17-19 UTC; Fig. 5a). The reduction of clouds at low levels is associated with a progressive deepening of the clouds (Fig. 5a), which simultaneously leads to an increase in precipitation (Fig. 5b). The deepest clouds associated with the highest rain rates are observed around 06 LT (10 UTC). This is in good agreement with Vial et al. (2019, their Fig. 3a,c), who also found a maximum in cloud fraction at low levels ($\sim 900\,\mathrm{m}$) during the night and at higher levels ($\sim 2500\,\mathrm{m}$) during early morning, as well as a peak in precipitation around 06 LT. In terms of absolute values, the rain rates in COSMO$_{\mathrm{iso,1km}}$ match the ones of Vial et al. (2019, ranging from 0.01 to $0.04\,\mathrm{mm\,h^{-1}}$). For cloud fractions, however, we find larger values at low levels and smaller values at higher levels compared to Vial et al. (2016). The underestimation of cloud fraction at higher levels was already noted and discussed in Villiger et al. (2023). The driver of this nighttime convective strengthening is assumed to be the horizontal inhomogeneity in the longwave cooling (Gray and Jacobson, 1977; Randall et al., 1991; Vial et al., 2019).

Convection, measured in terms of updraft strengths ($w_{up}$ in Fig. 5b), is strongest when clouds are deepest (Fig. 5a) and precipitation is most intense (Fig. 5b). In other words, the strongest updrafts occur around 06 LT (10 UTC) and the weakest ones around 18 LT (22 UTC). Vogel et al. (2020, their Fig. 3) and Vogel et al. (2022, their extended data Fig. 1) found similar diel cycles for the convective mass flux at cloud base in EUREC$^4$A observations, with high values during the morning and low values during the evening. Note, however, that the updraft strengths considered here are not directly comparable to the mass fluxes from Vogel et al. (2020, 2022). Their mass fluxes are about one order of magnitude smaller than our updraft strengths, because their definition takes into account the cloud-core area fraction and vertical velocity while we only consider vertical velocity.

Assuming an approximately closed cloud-relative circulation, we expect that the variation of the ascending branch (i.e., updraft strengths, rain rates, cloud fraction and vertical extent) over the day triggers a response in the descending branch. A physically meaningful response can be found in the COSMO$_{\mathrm{iso,5km}}$ trajectories: Air parcels arriving at *dry-warm* cloud base grid points require 10-14 h to cover the distance from the altitude where they were detrained from a cloud to the altitude of cloud base (red dashed line in Fig. 5c). In other words, the altitude at which these air parcels were detrained (red continuous line in Fig. 5c) should reflect the cloud characteristics 10-14 h earlier. If we, for example, look at the air parcels arriving 13 LT (17 UTC) at *dry-warm* cloud base grid points, we learn that these air parcels were detrained 12 h earlier (the time of the

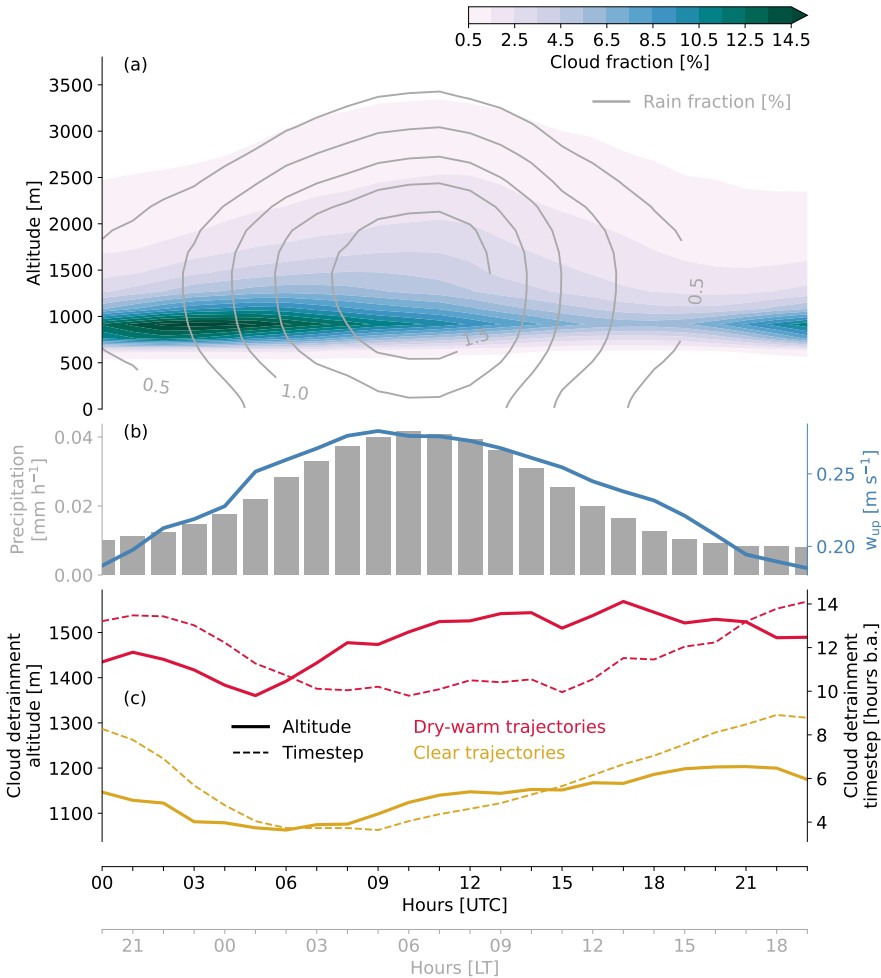

**Figure 5.** Diel cycle of **(a)** cloud (filled contours) and rain (contours) fraction at different levels, defined as fraction of grid points per model level exceeding the threshold of 10 and $1\,\mathrm{mg\,kg^{-1}}$ cloud and rain water, respectively; **(b, left)** average precipitation; **(b, right)** strength of updrafts at cloud base, defined as the mean of positive vertical velocities in *cloud* grid points ($w_{\mathrm{up}}$); and **(c)** median cloud detrainment altitude (continuous; left $y$-axis) and timestep (as hour before arrival; dashed; right $y$-axis) of the air parcels arriving at *clear* (yellow) and *dry-warm* (red) cloud base grid points, only considering those air parcel trajectories that have been inside a cloud (Sect. 2.4). The hour-of-the-day mean values in the domain 54.5-61° W and 11-16° N during the period 20 January to 13 February 2020 are shown. Data: **(a,b)** COSMO$_{\mathrm{iso,1km}}$ and **(c)** COSMO$_{\mathrm{iso,5km}}$ trajectories.

day with the highest low-level cloud fraction; Fig. 5a) from clouds at 1570 m. Contrastingly, the air parcels arriving 01 LT
(05 UTC) at *dry-warm* cloud base grid points were detrained 11 h earlier (the time of the day with the lowest low-level cloud
fraction; Fig. 5a) from clouds at 1360 m. Thus, the amount of low-level clouds and their vertical extent, determines the average
detrainment altitude of *dry-warm* air parcels.

A similar diel cycle is found for the air parcels arriving at *clear* grid points at cloud base. However, the subsidence times are
shorter (4-9 h; yellow dashed in Fig. 5c) and the cloud detrainment altitudes are correspondingly lower (1060-1200 m; yellow
continuous in Fig. 5c). Earlier (Sect. 2.3), we formulated the hypothesis that the *dry-warm* and *clear* cloud base characteristics
emerge due to different processes. The trajectory analysis discussed here suggests that it is more accurate to speak of different
paths than processes. The air forming the *dry-warm* patches originates from altitudes, where it is likely also mixed with air
from the free troposphere. Contrastingly, the *clear* environments form out of air that was detrained from clouds earlier, at
lower altitudes. In our understanding, both categories belong to the subsiding branch of the cloud-relative circulation, with the
*dry-warm* path connecting cloud base with higher detrainment altitudes.

If we link the two different transport histories with the idealised schematic, illustrating processes altering $\delta^2$H (Fig. 1a), we
come to the conclusion that we must find contrasting isotope signals in the three cloud base environments *dry-warm*, *clear*,
and *cloud*. The *dry-warm* air has ascended the furthest and should therefore be the most depleted environment. In addition,
mixing with free tropospheric air would further deplete the vapour before it subsides into the *dry-warm* patches at cloud base.
The *clear* air has ascended less and should therefore be more enriched. In the *cloud* air, condensation and rain out of heavy
isotopes just started, and therefore its vapour should be the most enriched.

The above considerations are confirmed by the data. Looking at $\delta^2$H (Fig. 6, left $y$-axis), we find the most enriched vapour
in the *cloud* and the most depleted vapour in the *dry-warm* cloud base environments. Similar characteristics are found for $q$
(Fig. 6, right $y$-axis), with the highest values in the *cloud* and the lowest in the *dry-warm* environments (as expected from the
definition of *dry-warm*). The three environments differ not only in absolute values, but also in the diel cycles. The diel cycles
of $\delta^2$H and $q$ are more pronounced for the *dry-warm* (amplitudes of $\sim$4‰ and 1.6 g kg$^{-1}$) and *clear* environments (amplitudes
of $\sim$1.6‰ and 0.8 g kg$^{-1}$) than for the *cloud* environments (amplitudes of $\sim$0.9‰ and 0.5 g kg$^{-1}$). The *cloud* patches are fed
by updrafts bringing moisture from the subcloud layer, in which the amplitude of the variability in $q$ and $\delta^2$H is small and of
about the same extent as observed in clouds (see part 1 of this study and Appendix A). While the cloudy grid points are thus
largely unaffected by the diel cycle of convection, the *dry-warm* and *clear* grid points reach their most depleted and driest state
shortly after noon local time, when cloud base cloudiness is minimal (Fig. 5a), updrafts are weak (Fig. 5b), and air parcels from
comparably high altitudes arrive in the two cloud base environments (Fig. 5c).

Two mechanisms likely contribute to the drying and depletion of the *clear* and *dry-warm* cloud base grid points over the
course of the day: (1) A small immediate effect due to the decrease in cloud fraction, which reduces the moistening and enrich-
ment of the *clear* and *dry-warm* patches through lateral detrainment from surrounding clouds. (2) A dominating temporally
delayed effect due to the vertical growth of clouds, which has the consequence that the detrainment from clouds happens at
increasing altitudes where temperature is lower and, therefore, saturated $q$ is less. With increasing altitudes, the isotope signal
is also more depleted due to the continuous condensation and rainout in the convective updrafts which transport the vapour

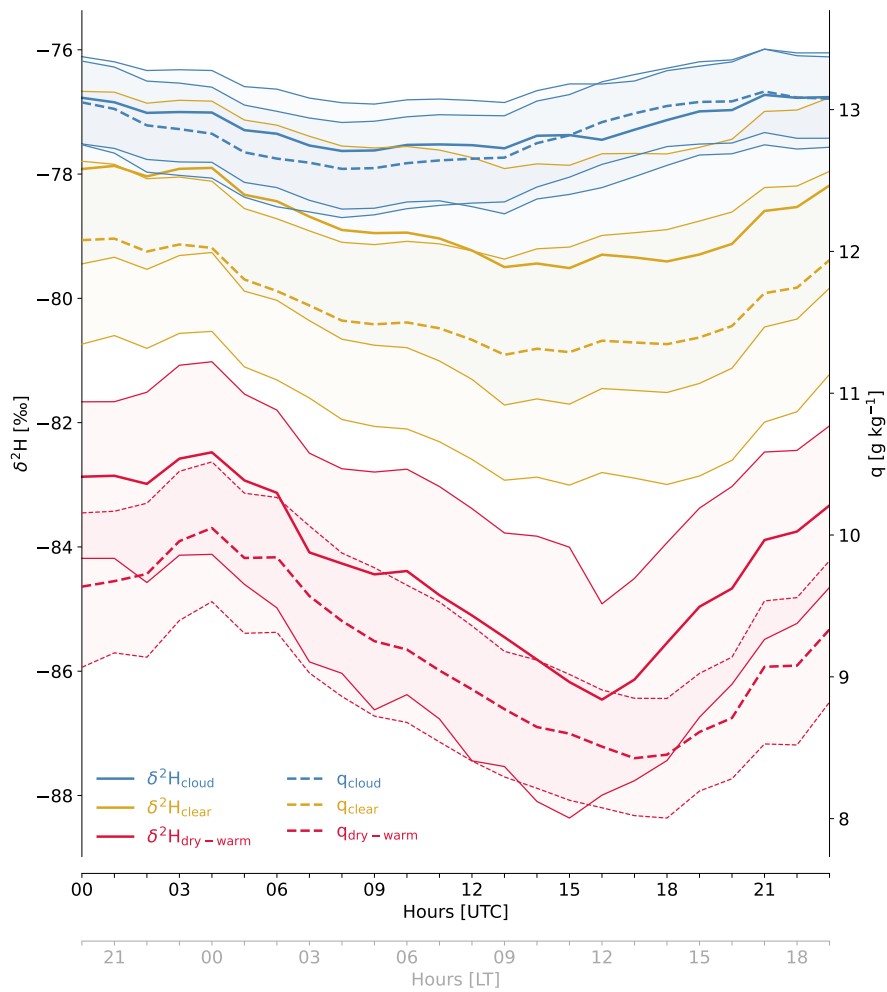

**Figure 6.** Diel cycle of $\delta^2$H in vapour with the median shown as a thick line and the 25-75-percentile range as shading/thin lines (continuous; left $y$-axis) and $q$ (dashed; right $y$-axis) of *cloud* (blue), *clear* (yellow) and *dry-warm* (red) cloud base grid points. The hour-of-the-day mean values in the domain 54.5-61° W and 11-16° N during the period 20 January to 13 February 2020 are shown. Data: COSMO$_{iso,1km}$.

to these altitudes in the first place. Assuming that the vapour detrained from clouds experiences little to no phase changes or mixing with advected vapour during its journey back to cloud base (i.e., closed circulation), it follows that its isotope signal is approximatively conserved. Thus, the higher the detrainment from clouds, the lower the amount of vapour that returns to cloud base, and the more depleted its isotope signal. This mechanism would explain the differences of $q$ and $\delta^2$H between the *clear* and *dry-warm* patches, as well as their diel cycles. A link between the altitude of origin (cf. detrainment altitude) and the depletion of the vapour was previously determined by Risi et al. (2019). Furthermore, George et al. (2023) discovered that the ascending branch of mesoscale circulations is associated with a moisture accumulation in the subcloud layer and at cloud base, while the descending branch is associated with a moisture deficit. Both studies, back our suggestion that the mesoscale spatial and temporal variability of cloud base isotopes is closely linked to the cloud-relative circulation.

As mentioned at the beginning of this section, we have to be careful with the absolute values of the variables derived from the COSMO$_{iso,5km}$ trajectories when combining them with variables from the COSMO$_{iso,1km}$ simulation. A comparison to literature, however, gives confidence that the obtained values for the detrainment time and altitude are meaningful. George et al. (2023) identified shallow mesoscale circulations in EUREC[4]A dropsonde observations as dipoles between the divergence in the subcloud layer and the divergence in the cloud layer. They find the highest lagged anti-correlation between the divergences of the two layers for a temporal lag of 7 to 8 h, which can be interpreted as a minimal lifetime of these circulations. This observation supports the subsidence times of 4 to 14 h as identified with our COSMO$_{iso,5km}$ trajectories. Furthermore, their definition of the cloud layer, ranging from 900 to 1500 m, largely overlaps with our diagnosed detrainment altitudes, giving them additional independent credibility.

In this section, we showed that at the subdaily timescale, $q$ and $\delta^2$H at cloud base vary little inside clouds, while their variations in the *clear* and *dry-warm* environments reflect the deepening of convection over the day, which (with a 4-9 h and a 10-14 h lag, respectively) causes drying and vapour depletion of the *clear* and *dry-warm* environments. The information gained from $q$ and $\delta^2$H seems congruent, while the d-excess contains no signal of a diel cycle in any cloud base environment (not shown; Villiger, 2022), pointing towards a dominant control by the large-scale circulation (see Sect. 4). Finally, we found evidence for the fact that the circulation associated with clouds is only a few hundred meters deep.

## 4 Cloud base isotopes and the large-scale circulation

This section investigates which of the three variables, $q$, $\delta^2$H, and d-excess, at the base of trade-wind clouds is most strongly influenced by the large-scale circulation. For this, we use trajectories calculated based on the COSMO$_{iso,10km}$ simulation (Sect. 2.5), which arrive evenly distributed near cloud base and are not targeted at *cloud*, *clear*, or *dry-warm* environments. We use the mean vertical displacement over time periods exceeding one day ($\overline{\Delta z}_{Xd}$) to distinguish between large-scale circulation patterns. For Hadley-cell-like subsidence, resulting from the balance between radiative cooling and adiabatic warming, a vertical displacement of $\sim 1.5\,\mathrm{hPa\,h^{-1}}$ (Salathé and Hartmann, 1997) is expected. For air parcels that go through an extratropical dry intrusion before arriving in the trades (Aemisegger et al., 2021; Villiger et al., 2022), the subsidence of individual air parcels exceeds $\sim 8\,\mathrm{hPa\,h^{-1}}$ (Raveh-Rubin, 2017).

**Table 1.** Pearson correlation coefficients between vapour $q$, $\delta^2$H, and d-excess of the COSMO$_{\text{iso,10km}}$ air parcels at their arrival near cloud base and their mean vertical displacement ($\overline{\Delta z}_{\text{Xd}}$; Sect. 2.5) during the X = 1 ... 6 d before arrival. The correlations are calculated between the mean values of the 138 air parcels arriving every hour between 22 January and 13 February 2020 (shown in Fig. 7). The strongest correlation for each variable is underlined. Combinations with no statistically significant association between the two variables (i.e., two-tailed $p$-value $> 0.05$; see $p$-values in Table B1) are highlighted in italics. Data: COSMO$_{\text{iso,10km}}$.

| | $\overline{\Delta z}_{\text{Xd}}$ | | | | | |
| --- | --- | --- | --- | --- | --- | --- |
| | 1d | 2d | 3d | 4d | 5d | 6d |
| $q$ | 0.29 | 0.33 | 0.19 | *0.05* | *0.07* | 0.13 |
| $\delta^2$H | 0.3 | 0.54 | 0.63 | 0.71 | 0.57 | 0.43 |
| d-excess | -0.16 | -0.25 | -0.37 | -0.54 | -0.72 | -0.75 |

The strongest link between the large-scale circulation and the cloud base $\delta^2$H in the trades is found if the air parcels' vertical displacement during 4 d before arrival is considered (Table 1). For the d-excess, the strongest link is found for 6 d or more (the trajectories' length is limited by the domain size in this analysis). The correlations are weaker for shorter or longer time scales, and the relationship is such that the vapour is more depleted and has a higher d-excess, the stronger the subsidence of the air parcels (Fig. 7b,c). In comparison to the isotope variables, $q$ is less influenced by the large-scale vertical displacement (comparably low correlations, with the maximum found for 2 days; Table 1 and Fig. 7a), illustrating the limited memory of moist processes registered by $q$ alone. This difference in memory of the history of moist processes along the flow in $q$ and $\delta^2$H is due to the fact that $q$ alone is determined to first order by the temperature just before the detrainment from clouds (i.e., last saturation paradigm, see Sherwood, 1996; Sherwood et al., 2010), while the $\delta^2$H contains information on both $[^1\text{H}_2{}^{16}\text{O}]$ and $[^1\text{H}^2\text{H}^{16}\text{O}]$, which relates to the condensation history in the clouds. The d-excess in turn connects to the non-equilibrium conditions at the evaporative moisture source, while being at first order unaffected by equilibrium cloud processing.

The physical link between the subsidence and $\delta^2$H is straightforward: the stronger the subsidence, the isotopically lighter the vapour because it originates from higher altitudes (Risi et al., 2019). The link between the subsidence and the d-excess can be explained by the contrasting conditions at the site where the vapour is evaporated and picked up by the air parcels embedded in the large-scale circulation (i.e., at the moisture source; see examples in Appendix C). Air parcels descending within an extratropical dry intrusion are expected to be drier and to create a stronger near-surface humidity gradient (i.e., lower RH$_{\text{SST}}$) at the moisture source (increasing the d-excess) than air parcels crossing the North Atlantic at low levels within the comparably moist trade winds where frequent detrainment of cloudy air occurs (lowering the d-excess; Fig. 1b). This interplay between large-scale subsidence, humidity gradient and d-excess is visualised in Fig. 8. The air parcels' d-excess at arrival is clearly more strongly influenced by the humidity gradient at the moisture source (Fig. 8a) than by the humidity gradient at their arrival location (Fig. 8b).

The maximum correlation shown in Table 1 is slightly higher for the d-excess than for the $\delta^2$H and small for $q$. Moreover, the vertical displacement time window leading to the highest correlation extends further back for the d-excess than for $\delta^2$H or

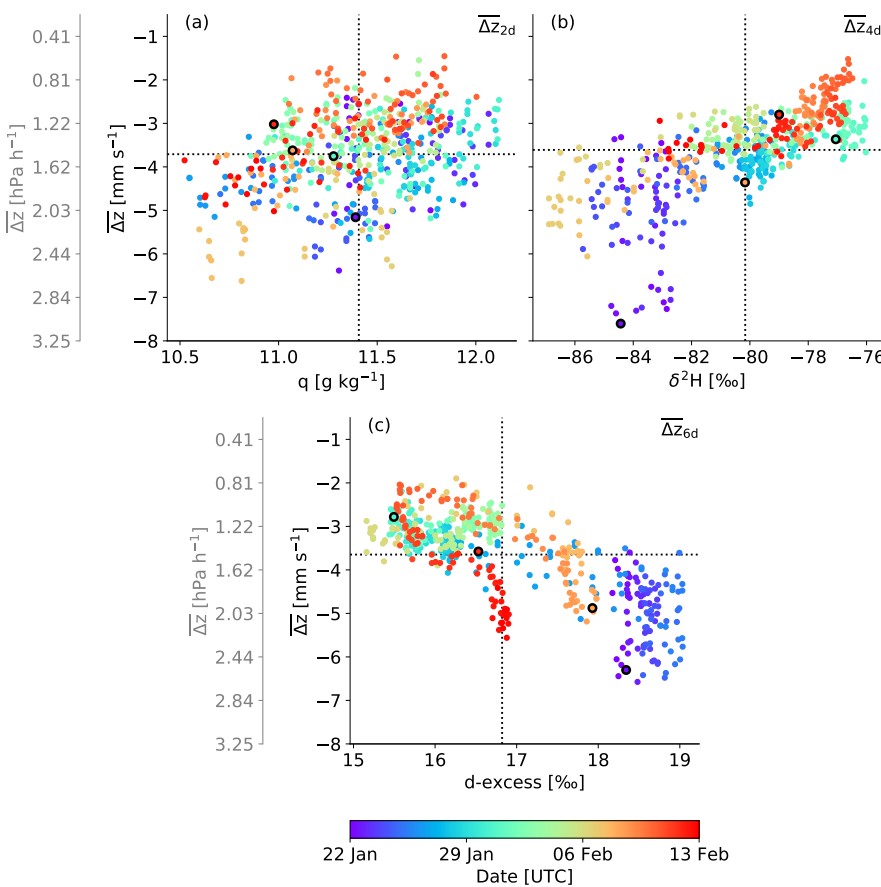

**Figure 7.** Relation between vapour **(a)** $q$, **(b)** $\delta^2$H, and **(c)** d-excess of the COSMO$_{iso,10km}$ air parcels at their arrival near cloud base and their mean vertical displacement ($\overline{\Delta z_{Xd}}$) over the period (X d), which yielded the highest correlation (Table 1), i.e., 2, 4, and 6 d. The data points are coloured according to the arrival timestep of the air parcels. Shown are the mean values over the 138 COSMO$_{iso,10km}$ air parcels arriving simultaneously (every hour from 22 January to 13 February 2020) at the three cloud base trajectory starting levels (940, 920, and 900 hPa) in the domain 54.5-61° W and 11-16° N. See Sect. 2.5 for details about how the subsidence is calculated. The dashed black lines indicate the mean values of the respective $x$- and $y$-variable over all arrival timesteps. The data points with black borders represent arrival at 12 UTC on 22 January 2020, 15 UTC on 31 January 2020, 18 UTC on 8 February 2020, and 18 UTC on 12 February 2020. Data: COSMO$_{iso,10km}$.

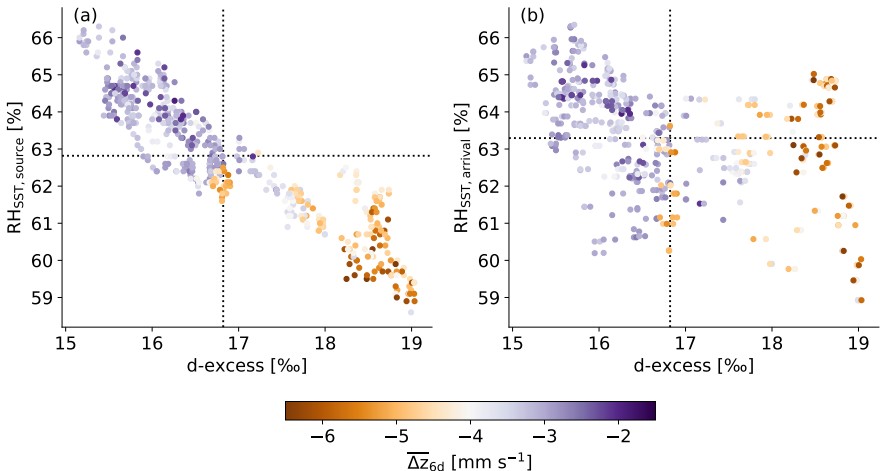

**Figure 8.** Relation between relative humidity with respect to sea surface temperature ($RH_{SST}$) and d-excess in vapour of the $COSMO_{iso,10km}$ air parcels at their arrival near cloud base. **(a)** $RH_{SST}$ at the moisture source of the air parcels (Pearson correlation coefficient= $-0.92$; two-tailed $p$-value $< 10^{-6}$); **(b)** $RH_{SST}$ at the air parcel's arrival location (Pearson correlation coefficient= $-0.37$; two-tailed $p$-value $< 10^{-6}$). The data points are coloured according to the mean vertical displacement of air parcels during the last 6 d before arrival at cloud base ($\overline{\Delta z_{6d}}$, i.e., the variable shown on the y-axis of Fig. 7c). Shown are the mean values over the 138 $COSMO_{iso,10km}$ air parcels arriving simultaneously (from 22 January to 13 February 2020) at the three cloud base trajectory arrival levels (940, 920, and 900 hPa) in the domain 54.5-61° W and 11-16° N. The dashed black lines indicate the mean values of the respective $x$- and $y$-variable over all arrival timesteps. Data: $COSMO_{iso,10km}$.

335 humidity. Both findings suggest that of the three variables, the d-excess is most strongly influenced by the large-scale circulation through the conditions created at the moisture source. This meets expectations, as the d-excess is known to be sensitive to the non-equilibrium fractionation conditions at the moisture source (Pfahl and Wernli, 2008; Aemisegger et al., 2021) and less to subsequent cloud processes, which often happen under conditions close to equilibrium. Contrastingly, $\delta^2H$ is sensitive to equilibrium and non-equilibrium processes, meaning the source signal is more quickly overwritten than the one of d-excess.

340 The mean vertical displacement over all arrival timesteps is $1.5\,\mathrm{hPa\,h^{-1}}$ (Fig. 7), which corresponds to a Hadley-cell-like descent. However, it can be substantially stronger or weaker for individual timesteps. To illustrate the variability in the large-scale circulation, four timesteps with contrasting 6-d subsidence are selected (Fig. 9). The two cases with weaker subsidence (Fig. 9a,b) are associated with the typical zonal flow of the trades. In the two cases with stronger subsidence (Fig. 9c,d), the trades are interrupted by Rossby-wave breaking events over the central North Atlantic, which steer the air parcels from

345 the extratropics towards low latitudes and cause a rapid descent along the slanted isentropes from the upper into the lower troposphere. As expected from the correlation analysis above (Table 1, Fig. 7), the different large-scale circulation patterns of the four cases lead to clear differences in the d-excess (ranging from 15.5 to 18.3‰) and the $\delta^2H$ (ranging from $-77.1$ to $-84.2$‰) at the air parcels' arrival location but not in $q$ (ranging from 11.0 to $11.4\,\mathrm{g\,kg^{-1}}$). This analysis on the influence of the large-scale circulation on cloud base isotopes showed, that on a daily timescale the d-excess is mainly impacted by the

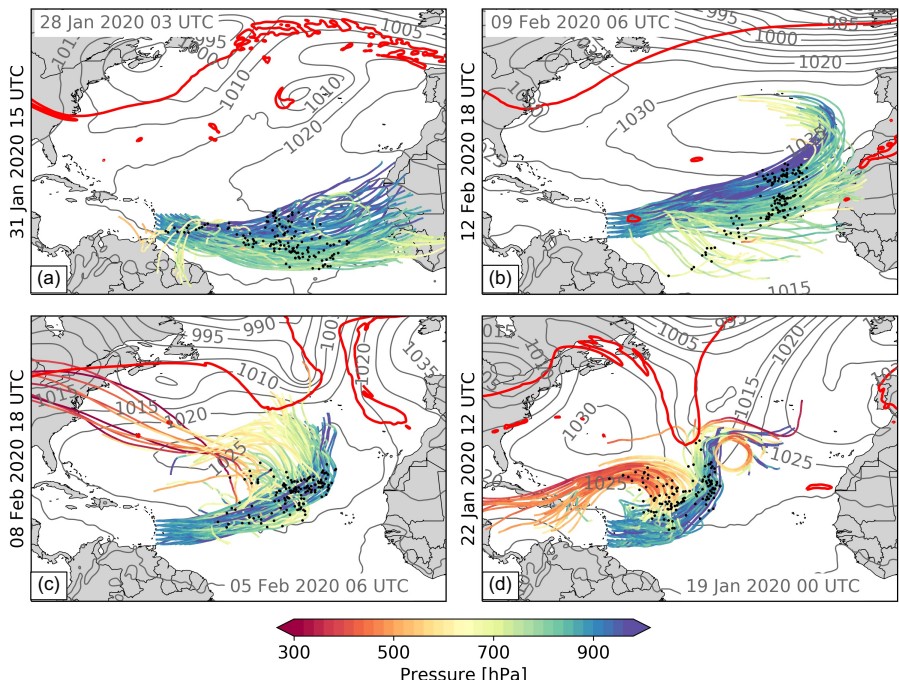

**Figure 9.** COSMO$_{iso,10km}$ trajectories for exemplary dates with different strength of the large-scale 6-d subsidence (the corresponding moisture sources are shown in Fig. C1). Arrival timesteps are indicated in black on the left side of the panels. The dates were selected from Fig. 7 as indicated by the black bordered markers. The trajectories arrive at 940, 920, and 900 hPa in the domain 54.5-61° W and 11-16° N. Together with each air parcel's position (black dots) 3.5 d before arrival, the sea level pressure (grey contours), and the dynamical tropopause (2 pvu at 320 K; red contours) are shown for the corresponding date indicated in grey at the top left/bottom right of each panel. Shown is the domain 0-85° W and 5° S-55° N and the air parcels arriving **(a)** at 15 UTC on 31 January ($q = 11.3$ g kg$^{-1}$, $\delta^2$H $= -77.1$‰, $d = 15.5$‰, 6-d subsidence $= -2.8$ mm s$^{-1}$ or 1.1 hPa h$^{-1}$); **(b)** at 18 UTC on 12 February ($q = 11.0$ g kg$^{-1}$, $\delta^2$H $= -79.0$‰, $d = 16.5$‰, 6-d subsidence $= -3.6$ mm s$^{-1}$ or 1.5 hPa h$^{-1}$), **(c)** at 18 UTC on 8 February ($q = 11.1$ g kg$^{-1}$, $\delta^2$H $= -80.2$‰, $d = 17.9$‰, 6-d subsidence $= -4.9$ mm s$^{-1}$ or 2 hPa h$^{-1}$), and **(d)** at 12 UTC on 22 January 2020 ($q = 11.4$ g kg$^{-1}$, $\delta^2$H $= -84.4$‰, $d = 18.3$‰, 6-d subsidence $= -6.3$ mm s$^{-1}$ or 2.6 hPa h$^{-1}$). The dates are sorted according to the strength of the subsidence over 6 d, starting with the case with the weakest subsidence in (a). Data: COSMO$_{iso,10km}$.

350    different regimes of the large-scale flow in the winter trades, while $\delta^2$H connects to the strength of the large-scale subsidence in addition to the strong influence of the cloud-relative circulation.

## 5 Summary and conclusion

In this final section, we combine the insights from relating isotope signals at cloud base to the circulation at different scales. In a first step (Sect. 3), we analysed the imprint of the cloud-relative circulation on cloud base water vapour isotopes in the trade-wind region. For this, we distinguished between three cloud base environments, *cloud*, *clear*, and *dry-warm*. We showed that the three environments differ regarding the values and diel cycles of $q$ and $\delta^2H$ and could attribute these differences to distinct processes in the cloud-relative circulation (see underlying concept in Fig. 1a). Furthermore, we demonstrated that $q$ and $\delta^2H$ of the *cloud* environment remain largely unaffected by the vertical growth of convection over the day, while the other two environments show a diel cycle linked to the cloud fraction at the cloud base and the cloud depth. Through the vertical growth of the clouds during the day, the altitude at which the vapour is detrained from the cloud is lifted. This cloud detrainment altitude of the air parcels arriving in the *clear* or *dry-warm* patches was identified as the main control of the degree of drying and depletion at cloud base. Since the cloud detrainment altitudes are higher for the *dry-warm* than for the *clear* environment, the drying and depletion is stronger in the former. The information gained from $q$ and $\delta^2H$ was largely congruent. However, by combining the two variables for the *dry-warm* environment (Fig. 10a), we find a group of data points (encircled in black) that is shifted towards lower $\delta^2H$ but shows no peculiarities in $q$ or the detrainment altitude. The stronger depletion of these data points can only be understood if the influence of the large-scale circulation is taken into account.

We investigated the imprint of the large-scale circulation on cloud base water vapour isotopes in a second step (Sect. 4). We illustrated that $\delta^2H$ and d-excess of the vapour near the cloud base (independently of the cloud base environment, *cloud*, *clear*, or *dry-warm*), but not $q$, vary on the synoptic scale following changes in the large-scale circulation in a physically meaningful way. Namely, the vapour in the trade-wind region is more depleted and has a higher d-excess the stronger the subsidence (i.e., the more negative the vertical displacement) during the preceding 4 to 6 d. The $\delta^2H$ in cloud base vapour was found to be most strongly linked to the 4-d vertical displacement (Pearson correlation of $0.71$), which also explains the more depleted data points (encircled in black) in Fig. 10. Contrastingly, the d-excess is most strongly linked to the 6-d vertical displacement (Pearson correlation of $-0.75$). The strong link between the d-excess and the large-scale circulation found here is in agreement with Aemisegger et al. (2021, their Fig. 16), who found a clear link (Pearson correlation of $-0.73$) between the d-excess in vapour measured on Barbados in the subcloud layer and the distance to the moisture source, during a field experiment in early 2018. The analysis also showed that the identified pathways of air parcels arriving in the lower troposphere qualitatively agree with the ones identified in Villiger et al. (2022).

In Fig. 10, we summarize the main findings of this study. Namely, the values of $q$ and $\delta^2H$ in *dry-warm* cloud base environments are determined (a) by the cloud-relative circulation through the cloud detrainment altitude (reflecting the diel cycle of convection) as well as (b) by the large-scale subsidence reflecting different circulation patterns with distinct moisture source conditions. The isotope-shaping processes within these two circulations occur at different timescales. While the cloud detrainment takes place 4 to 14 h before the air parcels arrive at cloud base, the moisture source conditions are shaped by the subsidence during the 4 to 6 d before the air parcels arrive in the trades. We, here, focus on the *dry-warm* environments because it is the cloud base environment where the influence from the cloud-relative circulation is strongest.

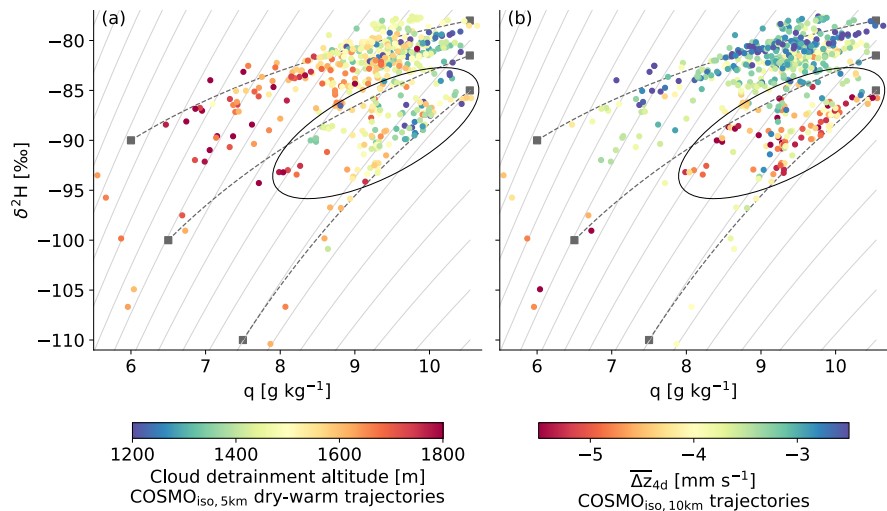

**Figure 10.** Relation between hourly median values of $q$ and $\delta^2$H in the vapour of the cloud base grid points identified as *dry-warm* in the domain 54.5-61° W and 11-16° N from COSMO$_{iso,1km}$. They are coloured according to **(a)** the cloud detrainment altitude derived from the COSMO$_{iso,5km}$ trajectories, and **(b)** $\overline{\Delta z}$ over 4 d from the COSMO$_{iso,10km}$ trajectories. Rayleigh distillation curves for an open system (i.e., assuming 100 % precipitation efficiency; light gray continuous) and three exemplary mixing lines (dark gray dashed) with the mixing end members (dark gray squares) are shown for reference. Data points with enhanced large-scale subsidence leading to low $\delta^2$H values are encircled in black.

To deepen our understanding of the hydrological cycle associated with shallow trade-wind cumulus clouds, the role of mixing and liquid-vapour interaction processes embedded in the cloud-relative circulation should be examined in more detail. In Fig. 10 the $\delta^2$H-$q$ value pairs of *dry-warm* cloud base environments together with reference lines illustrating the impact of mixing (dark gray dashed lines) and precipitation production in clouds (assuming 100 % precipitation efficiency; light gray

lines) show that a combination of processes is involved including mixing and microphysical processes. To disentangle the influence of these processes on the isotope signal, the cloudy profiles from EUREC[4]A observations and simulations should be studied in more detail in the future. Furthermore, a tagging experiment with numerical tracers distinguishing subcloud-layer and free-tropospheric moisture would help to disentangle the contribution of the circulations associated with clouds of varying depths and the large-scale circulation (Brient et al., 2019). Our analysis demonstrated that isotopes represent the integral signal

of past moist atmospheric processes encountered along the flow. Particularly novel was the finding that isotopes serve as indicators for changes in atmospheric circulations on various scales. By investigating the isotope signal at cloud base in the trade-wind region, we could identify the imprint of different large-scale circulation patterns and the circulation associated with clouds that, in concert, determine the formation of clouds and precipitation.

## Appendix A: Evaluation of the simulated diel cycle with observations

In this appendix, the modelled diel cycles (Fig. A2) of near-surface humidity and isotope variables are evaluated using observations (Bailey et al., 2023) from the EUREC[4]A field campaign (Stevens et al., 2021). For this purpose, the data points from the lowest model level and the grid point closest to the Barbados Cloud Observatory (BCO; Fig. A1) were extracted from the three COSMO$_{iso}$ simulations. Note that an evaluation of the diel cycles of cloud base variables is not possible since measurements at this altitude were restricted to flight hours, which do not encompass the complete 24 h day.

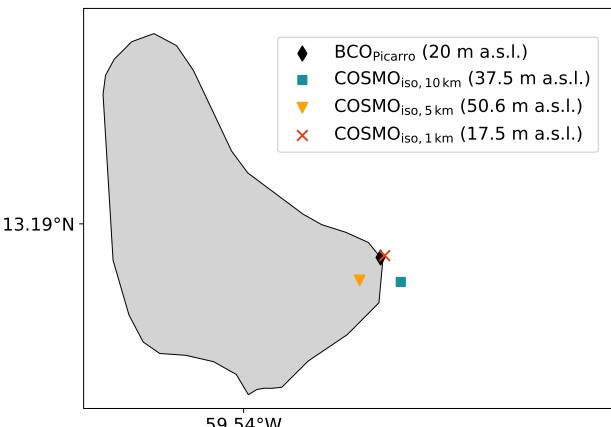

**Figure A1.** Grid point of the COSMO$_{iso,10km}$ (teal), COSMO$_{iso,5km}$ (yellow), and COSMO$_{iso,1km}$ (red) model setup closest to the BCO (black marker). The altitude of the lowest model level is given in brackets (in meter above sea level [m a.s.l.]. The BCO is located 17 m a.s.l. (Stevens et al., 2016, their Fig. 2) and the isotope observations were conducted at $\sim$ 3 m above ground (Bailey et al., 2023).

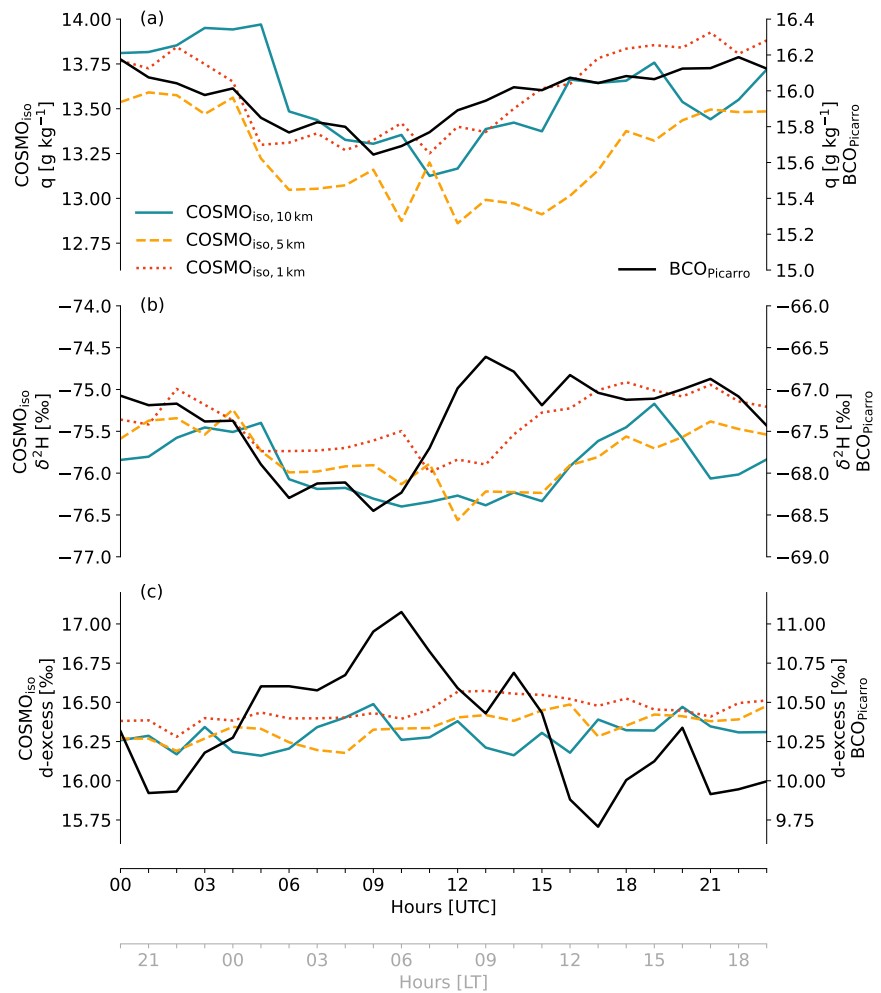

**Figure A2.** Diel cycles of near-surface **(a)** $q$, **(b)** $\delta^2H$, and **(c)** d-excess in the COSMO$_{iso,10km}$ (teal), COSMO$_{iso,5km}$ (yellow dashed), and COSMO$_{iso,1km}$ (red dotted) simulations together with observations collected during EUREC[4]A at the Barbados Cloud Observatory (BCO; black, labelled BCO$_{Picarro}$; see Bailey et al., 2023, for a description of the BCO observations). For all four datasets, the period from 20 January to 13 February 2020 is considered for the calculation of the diel cycle. For COSMO$_{iso}$, the data from the lowest model level and the grid point closest to the BCO is selected (Fig. A1). Note that the lowest model level is different for the three COSMO$_{iso}$ setups and does not correspond to the altitude of the observations. Therefore, different $y$-axes are used for the simulations (left) and the observations (right).

# Appendix B: Relation between cloud base conditions and large-scale circulation

In this appendix, two-sides $p$-values (Table B1) for the correlations listed in Table 1 are given. They indicate that all correlations are statistically significant, except for the one between cloud base $q$ and the large-scale subsidence over $4\,\mathrm{d}$ ($\overline{\Delta z_{4\mathrm{d}}}$) or $5\,\mathrm{d}$ ($\overline{\Delta z_{5\mathrm{d}}}$; see Table B1).

**Table B1.** Two-sided $p$-values indicating whether there is a statistically significant association between two variables. The same variable combinations as in Table 1 are investigated. Non-significant associations between two variables (i.e., two-tailed $p$-value $> 0.05$) are highlighted by underlining. Data: $COSMO_{\mathrm{iso,10km}}$.

| | $\overline{\Delta z_{\mathrm{Xd}}}$ | | | | | |
|---|---|---|---|---|---|---|
| | 1d | 2d | 3d | 4d | 5d | 6d |
| $q$ | $1.9 \times 10^{-12}$ | $3.2 \times 10^{-15}$ | $4.2 \times 10^{-6}$ | $\underline{2.8 \times 10^{-1}}$ | $\underline{9.7 \times 10^{-2}}$ | $2.1 \times 10^{-3}$ |
| $\delta^2\mathrm{H}$ | $1.0 \times 10^{-12}$ | $2.0 \times 10^{-43}$ | $2.5 \times 10^{-61}$ | $3.4 \times 10^{-85}$ | $1.7 \times 10^{-49}$ | $1.1 \times 10^{-25}$ |
| d-excess | $2.4 \times 10^{-4}$ | $4.6 \times 10^{-9}$ | $3.1 \times 10^{-19}$ | $9.0 \times 10^{-43}$ | $2.4 \times 10^{-89}$ | $1.8 \times 10^{-100}$ |

## Appendix C: Moisture uptake along the COSMO$_{iso,10km}$ trajectories

In this appendix, the moisture uptakes along the trajectories shown in Fig. 9 are displayed (i.e., the moistures sources).

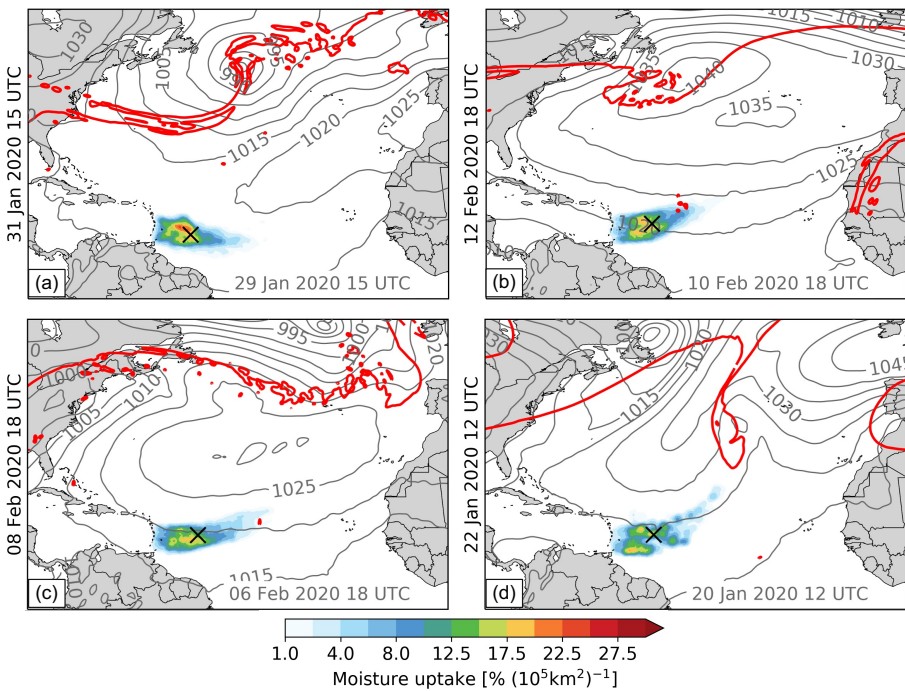

**Figure C1.** COSMO$_{iso,10km}$ moisture sources for the trajectories shown in Fig. 9. The arrival timesteps, indicated in black on the left side of the panels, are **(a)** 15 UTC on 31 January, **(b)** 18 UTC on 12 February, **(c)** 18 UTC on 8 February, and **(d)** 12 UTC on 22 January 2020. The sea level pressure (grey contours) and the dynamical tropopause (2 pvu at 320 K; red contours) are shown 2 d before arrival. The corresponding date is indicated in grey at the top left/bottom right of each panel. Shown is the domain 0-85° W and 5° S-55° N. Data: COSMO$_{iso,10km}$.

*Data availability.* The COSMO$_{iso}$ simulations are published in the ETH research collection (Villiger and Aemisegger, 2022).

*Author contributions.* FA designed the project and acquired the funding. LV with the support of FA carried out the COSMO$_{iso}$ simulations, performed the data analysis, and wrote the paper. LV and FA discussed the results and the structure of the paper in detail.

*Competing interests.* The authors have no competing interests to declare.

*Acknowledgements.* We thank Heini Wernli (ETH Zürich) for the scientific advice and support in this project as well as for his comments on the manuscript. We appreciate the technical support from Urs Beyerle (ETH Zürich) regarding the cluster on which the model calculations were performed and the help from Fabienne Dahinden with the installation of the model on the cluster. We are grateful to "La Fondation des Treilles" for hosting the workshop "How and why do shallow clouds organise? First lessons from EUREC$^4$A" (21 – 26 Mar 2022) organised by Sandrine Bony and Bjorn Stevens. This workshop served as a source of inspiration for this paper. We are grateful to Ann Kristin Naumann

and an anonymous reviewer for their insightful and constructive comments, which helped to improve the clarity of the manuscript.

*Financial support.* LV received funding from the Swiss National Science Foundation (grant no. 188731).

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
