# Peer review of "Water isotopic characterisation of the cloud-circulation coupling in the North Atlantic trades. Part 2: The imprint of the atmospheric circulation at different scales"

_EGUsphere, 2023_

## Referee Comment (RC1)

**Review of Villiger and Aemisegger, part 2**

April 17, 2023

This article investigates the relative importance of the circulations at the mesoscale and at the large scale in controlling the temporal and spatial water vapor isotopic variability in trade wind regions. It contrasts the relative impact of circulations at different scales on humidity, $\delta^2 H$ and second-order parameter d-excess. It relies on high-resolution isotope-enabled simulations with COSMO-iso, which were thoroughly evaluated in part 1. This article is very interesting. I think it is relevant for the water isotope community, but also for the cloud-process community in trade wind regions.

The article is well written and well illustrated. However, while I found part 1 too long, I found this paper too short. I have a few major comments.

**1   Major comments**

- **What is the realism of the simulated diel cycle of cloud fraction, cloud depth and water isotopes, with respect to observations?**
  I don't remember that this was evaluated in part 1. In particular the amplitude of the diel cycles and the relative phasing for different variables were not evaluated.
  Is there enough observations to evaluate the simulated diel cycle? If not, how risky is it to base all the discussion on the role of mesoscale circulation on the diel cycle?
  Isn't there a way to investigate the role of mesoscale circulation without relying on the diel cycle? e.g. by looking directly at dry-warm patches anomalies, independantly of the diel cycle?

- **Discussion in section 3 is too short**
  I think Fig 5 leads to plenty of questions and I found it very stimulating, so I was quite frustrated at the end of section 3 that the discussion was so short and that I was left with so many unanswered questions. For example:

  - is the delay of the $\Delta z$ for 12h relative to 1h just the impact of the averaging back in time?
  - why is the $\delta^2 H$ minimum in dry-warm patches delayed relative to the precipitation maximum and to the maximum subsidence? Is it because (1) $\delta^2 H$ has some memory as it integrates processes back in time? Or is it because (2) it instantaneously adapts to some other variable that is delayed relative to the precipitation maximum, e.g. the cloud-base cloud fraction? In case of the (2), what would be the mechanism?
  - why is the cloud-base cloud fraction minimum delayed relative to the precipitation maximum? l 129-130 is not convincing because cloud fraction does not look in phase with cloud depth.

- **Relative impact of local processes and large-scale circulation on d-excess?**

  - l 177-178: "humidity gradient": where? locally, or at the moisture source? If dry intrusions are associated with a drier near-surface air locally and thus higher d-excess in the surface evaporation flux locally, we would expect a good correlations between d-excess and surface relative humidity. Is it the case?
  - l 181: "influenced by the large-scale cirulation": same question as above: is it really the large-scale circulation that directly impact d-excess? Or is it the local surface relative humidity, impacted by the large-scale circulation, that impacts d-excess?

**2  Minor comments**

- l 6: what is the subject of "show"? Cut this sentence into 2 sentences for clarity?

- l 126-127: this technical comment could go in the Methods section, in 2.4.

- Fig 5: I find it hard to see the relative proportions of shallow, middle and deep clouds. They all look flat. Is it possible to rather show the proportion of deep cloud as a line, like for the cloud fraction?

- l 136: why are the variations in cloudy patches small? Is it because the vapor comes from the near-surface, where the temporal variations are small?

- l 146: "subsidence rate": is it really a rate, or do you mean the altitude of origin?
  Same question l 200: "subsidence ... stronger": is the subsidence stronger or originating from a higher altitude? Or is it equivalent? (and if so, what process links the subsidence velocity with the origin altitude of air parcels?)

- l 167: we don't know for d-excess: is it possible to extend the duration of the trajectories?
  I would make 2 separate sentences for $\delta^2 H$ and d-excess, since for $\delta^2 H$ the maximum link is for 4 days, whereas for d-excess it is for 6 days or larger, we don't know.

- l 170: "not" -> "less"

- Table 1: is it possible to give some p-values associated with the correlation coefficients, or some correlation coefficient threshld to reach a given p-value?

- l 184: this is interesting. Can you also elaborate on what are the mechanisms that overwrite $q$ faster than $\delta^2 H$?

- l 218-219: This idea is very interesting. This recalls me the study of [Brient et al., 2019] who identifies coherent structures in shallow clouds. It would be very interesting to look at the isotopic signature of these different sructures. Maybe this study could be cited.

- Fig 8: it would be useful to add a Rayleigh curve and a mixing curve to guide the interpretation of these plots.

**References**

[Brient et al., 2019] Brient, F., Couvreux, F., Najda, V., Rio, C., and Honnert, R. (2019). Object-oriented identification of coherent structures in large-eddy simulations: importance of downdrafts in stratocumulus. *Geophy. Res. Lett.*, 46:2854–2864, https://doi.org/10.1029/2018GL081499.

---

## Author Comment (AC1)

**Authors' response to reviews of the manuscript**

**"Water isotopic characterisation of the cloud-circulation coupling in the North Atlantic trades. Part 2: The imprint of the atmospheric circulation at different scales"**

by Leonie Villiger and Franziska Aemisegger

We thank the anonymous reviewer (hereafter reviewer 1) and Ann Kristin Naumann (hereafter reviewer 2) for their insightful comments, which we address in detail in our responses below. They helped us to improve the presentation of our results and will lead to the following main changes in the revised manuscript:

1. We will include a short evaluation of the diel cycles in vapour isotopes in the Appendix using observations from a near-surface coastal site as a response to **reviewer 1 major point A and reviewer 2 major point B**.

2. We will add more background information about isotopes in the introduction, as well as more details about the COSMO$_{\text{iso}}$ simulations in Sect. 2.1 (Applied datasets) as a response to **reviewer 2 major point A and comment 2.5**.

3. We will extend the discussion and improve the presentation of the diel cycle in Sect. 3 (Cloud base isotopes and the cloud-relative circulation) as a response to **reviewer 1 major point B and reviewer 2 comments 2.7, 2.13, and 2.15**.

4. We will add a short analysis in Sect. 4 (Cloud base isotopes and the large-scale circulation) showing that the d-excess of vapour is more strongly influenced by the humidity gradient at the moisture source than by the humidity gradient at a given point of observation as a response to **reviewer 1 major point C**.

Please find our detailed response to the reviewers' comments below in the following format:
Reviewer's comment. Authors' response.
* * *
**Reviewer 1:**

**General comment**: The article is well written and well illustrated. However, while I found part 1 too long, I found this paper too short.
We thank the reviewer very much for her/his constructive and insightful comments, which have helped us to improve the clarity of the manuscript. The first submitted version of this paper was very short because it was initially planned as a letter. We are currently adapting the manuscript providing more detail in the text, where the reviewers asked for more discussion. We will also implement additional analyses (in particular on the diel cycle), which aim at facilitating the understanding of the manuscript.

**Major comments**:

A **What is the realism of the simulated diel cycle of cloud fraction, cloud depth and water isotopes, with respect to observations?** I don't remember that this was evaluated in part 1. In particular, the amplitude of the diel cycles and the relative phasing for different variables were not evaluated.
In part 1, we did evaluate many aspects of isotope signals associated with shallow cumulus cloud formation and their environment, but not the daily cycle because observations of the daily cycle at cloud base are not available and because we wanted to treat this aspect in part 2.

A.1 Is there enough observations to evaluate the simulated diel cycle?
Here we split our response in a discussion of non-isotope and isotope variables.

**Non-isotope variables**: The diel cycle of cloud fraction, cloud depth, precipitation, and convective mass flux at cloud base (to some extent comparable to the updraft strength considered in Fig. 5) has been evaluated in detail in previous publications and the characteristics of the diel cycle found here in COSMO$_{\text{iso}}$ are very close to the findings in Vial et al. (2019) and Vogel et al. (2020,2022), in which Large-Eddy simulations with ICON and have been combined with observations. We will add references to these studies in the manuscript as described in the response to the reviewer comment 2.16.

**Isotope variables**: Robust observations of the diel cycle of isotope signals at cloud base are very difficult to obtain. We have aircraft observations from different times over the day but despite the considerable deployment effort, we do not have enough data to cover the whole diel cycle at the cloud base level and for the different cloud base features in a robust way (see Fig. R1). In qualitative terms, the observations we have from the ATR point towards a minimum in $\delta^2$H in the morning hours, which is delayed compared to the minimum in $q$. But this is very speculative given the 18 available four-hour flights spread between 3 and 18 LT.

We do have ground-based observations from the Barbados Cloud Observatory (BCO), which we show in Fig. R2 together with the lowest model level data from the three COSMO$_{\text{iso}}$ simulations using the closest grid points (see Fig. R3). The diel cycles in isotope signals near the surface and at cloud base (between 780 and 970 m a.s.l.) do not necessarily have to be synchronous or of the same amplitude, because different processes may dominate at the two levels:

- stronger and more immediate influence of surface evaporation for the near-surface BCO data
- potential impact of land surface processes for the BCO site compared to the free ocean location at which the daily cycle at cloud base is analysed.

When comparing the simulation data in Fig. R2a,b (diel cycle of near-surface BCO $q$ and $\delta^2$H) to the cloudy group in the new Fig. 6 (former Fig. 5) of the manuscript (diel cycle of cloud base $q$ and $\delta^2$H in clouds), we observe very similar timings of minima (3-6 LT) and maxima (15-21 LT) in $q$ and $\delta^2$H. This is plausible, since the clouds are feed primarily by humidity from the mixed layer and thus near-surface isotope and humidity signals are most likely to match with cloudy signals at cloud base. We observe a delayed diel cycle at cloud base compared to the near-surface BCO site, when comparing the BCO data to the clear sky and dry-warm patches: the maximum at BCO occurs earlier in the day (12-21 LT) in all three simulations, with a double peak in the 10 km simulation (15 LT and 2 LT). This delay is most likely due to the turnover time of the shallow circulation. Surface evaporation more strongly and more immediately impacts the near-surface BCO site. Furthermore, mixed layer isotope signals are affected by below cloud interaction with the falling rain. The amplitude of the diel cycle at cloud base is slightly larger for both $q$ and $\delta^2$H compared to the BCO, most likely owing to the stronger direct influence of dry and depleted air subsiding from cloud top regions and the much weaker impact of turbulent upward mixing of moisture from fresh ocean evaporation.

For the BCO, we see a relatively good agreement between the observed and the simulated diel cycles, both in terms of amplitude (Table R1) and phasing (Fig. R2). Shortcomings in the model are linked to (1) a too small amplitude of the diel cycle in the simulated $d$-excess (in agreement with the underestimation of the observed variability in $d$-excess by the COSMO$_{\text{iso}}$ simulations discussed in part 1) and (2) a slightly too late increase in $\delta^2$H in the morning hours. From this short analysis including the BCO data, we conclude that the evaluation of the diel cycle with the existing observations points to an adequate representation of the physical ingredients shaping the diel cycle in the model.

We would like to emphasise here, that despite the fact that we cannot evaluate the diel cycle of isotopes in the model in detail with direct observations, we have evaluated many variables connected to it in a physical way (such as cloud base cloud fraction, precipitation isotope signals at cloud base and in the mixed layer; see Villiger, 2022).

We will add a small note about this short evaluation of the daily cycle using the BCO observations in the manuscript at the beginning of Sect. 3 and include Fig. R2 and Fig. R3 in the appendix of the manuscript.

[Figure]

**Figure R1:** Diel cycles of cloud base $\delta^2H$ (left axis) and specific humidity (right axis) from COSMO$_{iso,1km}$ in different cloud base environments (as in Fig. 5 [now 6] of the paper). Median (marker) and the 25-75-percentile range (transparent line/errorbars) of cloud base $\delta^2H$ (left axis) and specific humidity (right axis) for each ATR flight with isotope observations (RF03-RF20). Refer to Bailey et al. (2023) for a description of the ATR isotope observations.

[Figure]

**Figure R2:** Diel cycles of near-surface (a) specific humidity, (b) $\delta^2$H, and (c) d-excess in the COSMO$_{iso,10km}$ (teal), COSMO$_{iso,tkm}$ (yellow), and COSMO$_{iso,1km}$ (red) simulations together with observations collected during EUREC$^4$A at the Barbados Cloud Observatory (BCO; black, labelled BCO$_{Picarro}$); see Bailey et al. (2023) for a description of the BCO observations). For all four datasets, the period from 20 January to 13 February 2020 is considered for the calculation of the diel cycle. For COSMO$_{iso}$, the data from the lowest model level and the grid point closest to the BCO is selected (Fig. R3). Note that the lowest model level is different for the three COSMO$_{iso}$ setups and does not correspond to the altitude of the observations. Therefore, different y-axes are used for the simulations (left) and the observations (right).

[Figure]

**Figure R3:** Grid point of the $COSMO_{iso,10km}$ (teal), $COSMO_{iso,5km}$ (yellow), and $COSMO_{iso,1km}$ (red) model setup closest to the BCO (black marker). The altitude of the lowest model level is given in brackets (in meter above sea level [m a.s.l.]. The BCO is located 17 masl (Stevens et al., 2016, their Fig. 2) and the isotope observations were conducted at $\sim 3$ m above ground (Bailey et al., 2023)

**Table R1:** Amplitude (maximum - minimum) of the diel cycles shown in Fig. R2 R2.

|  | $COSMO_{iso,\ 10km}$ | $COSMO_{iso,\ 5km}$ | $COSMO_{iso,\ 5km}$ | $BCO_{Picarro}$ |
|---|---|---|---|---|
| $q$ | 0.84 | 0.73 | 0.67 | 0.54 |
| $\delta^2H$ | 1.23 | 1.32 | 1.08 | 1.84 |
| d-excess | 0.33 | 0.31 | 0.29 | 1.37 |

A.2 If not, how risky is it to base all the discussion on the role of mesoscale circulation on the diel cycle?
Radiative forcing and the resulting diurnal cycle is a first order control on the dynamics of shallow cumulus clouds. We therefore argue that by studying mesoscale circulations without taking the diel cycle into account, we would neglect a key driver of variability in the trade wind cloudiness. Furthermore, the focus on the diel cycle provides a useful and simple framework to study the physical mechanisms driving shallow convection. Finally, with the studies by Vial et al. (2019) and Vogel et al. (2020) there is already a solid literature basis on the diurnal cycle of shallow cumulus clouds in the western tropical North Atlantic, on which we can build our analysis that introduces isotopes as integral tracers of moist processes.

A.3 Isn't there a way to investigate the role of mesoscale circulations without relying on the diel cycle? e.g. by looking directly at dry-warm patches anomalies, independently of the diel cycle?
We do indeed have several ideas for new investigations on dry-warm patches in future research, but we argue that the radiative forcing driving the strength of mesoscale circulations is a first order control on isotope signals associated with trade wind cumulus cloud formation that should be investigated first.

B **Discussion in section 3 is too short.** I think Fig 5 (now split into several figures) leads to plenty of questions and I found it very stimulating, so I was quite frustrated at the end of section 3 that the discussion was so short and that I was left with so many unanswered questions.
We understand this frustration. This was due to our initial choice of keeping the manuscript in a very compact letter format. We will extend the discussion of this part and also add new analyses (also as a response to the reviewer comments 2.7, 2.13, and 2.15). In particular, we will (1) also discuss the diel cycle of the clear-sky cloud base environments (complementary to the cloudy and dry-warm cloud base patches), (2) we will also calculate and analyse $COSMO_{iso,5km}$ trajectories from the clear-sky cloud base environments, and (3) we will replace the subsidence analysis ($\overline{\Delta z}$ over different time windows) of the $COSMO_{iso,5km}$ trajectories with an analysis of the point of last saturation (i.e., the time step and altitude of the air parcel when it last was inside a cloud).

For example:

B.1 Is the delay of the $\Delta z$ for 12h relative to 1h just the impact of the averaging back in time?

Yes, the delay of the $\Delta z$ for 12h relative to 1h is the impact of averaging over a longer time period backward in time. This effect is illustrated in Fig. R4, which shows the diel cycle of $\Delta z$ over different time windows. $\Delta z$ for 1 h, for example, only informs about the vertical displacement that the air parcels experienced during the last hour before they arrive at cloud base (i.e., provides an almost instantaneous description of the air parcels' subsidence). This 1 h-subsidence is strongest for the air parcels arriving at cloud base at approx. 4 LT. However, the 1 h-subsidence does not inform about the vertical displacement that the air parcels underwent earlier in their journey. Considering the vertical displacement over longer time periods (i.e., following the air parcels further back in time) gives a more robust estimate of the overall subsidence of the air parcels.

Note, however, that we will replace the analysis of $\Delta z$ of the $COSMO_{iso,5km}$ trajectories with another approach. Therefore, this comment will become irrelevant for the revised version of the manuscript.

[Figure]

**Figure R4:** Diel cycle of $\overline{\Delta z}$ over different time windows (following the air parcel trajectories 1 to 24 h backward in time) derived from the $COSMO_{iso,5km}$ trajectories. Similar to Fig. 5b in the original version of the paper.

B.2 Why is the $\delta^2 H$ minimum in dry-warm patches delayed relative to the precipitation maximum and to the maximum subsidence? Is it because (1) $\delta^2 H$ has some memory as it integrates processes back in time? Or is it because (2) it instantaneously adapts to some other variable that is delayed relative to the precipitation maximum, e.g. the cloud-base cloud fraction? In case of the (2), what would be the mechanism?

We strongly argue for (1): the isotope signal of the dry-warm patches is shaped primarily in the cloudy updrafts where condensation takes place. During subsequent subsidence in the dry-warm blobs, the isotope signal is conserved while being transported downward. The delay between the precipitation maximum and the $\delta^2 H$ minimum in dry-warm patches is therefore due to the transport time from the updraft, where precipitation forms and, where the condensational depletion happens until the arrival of the vapour and its depleted isotope signal back at cloud base. The delay between the $\delta^2 H$ minimum in dry-warm patches and the 1-h subsidence is due to the turnover time of the mesoscale shallow circulations. The $\delta^2 H$ diel cycle is in phase with the 12h subsidence rates, indicating that the turnover times must be in this time range.

About process (2) suggested by the reviewer: The fact that the diel cycle of $\delta^2 H$ in dry-warm patches is in phase with the cloud fraction is due to an indirect effect. Cloud fraction does not directly control $\delta^2 H$ in dry warm patches. But cloud fraction is minimum when the subsidence strength integrated over the turnover times of the mesoscale circulations is maximum. We will clarify these points in Section 3.

B.3 Why is the cloud-base cloud fraction minimum delayed relative to the precipitation maximum? L129-130 is not convincing because cloud fraction does not look in phase with cloud depth.

This is due to the subsidence-delay discussed above. We will clarify these points in Section 3.

**C Relative impact of local processes and large-scale circulation on d-excess?**

C.1 L177-178: "humidity gradient": where? locally, or at the moisture source? If dry intrusions are associated with a drier near-surface air locally and thus higher d-excess in the surface evaporation flux locally, we would expect a good correlations between d-excess and surface relative humidity. Is it the case?

We meant the humidity gradient at the moisture source. We have adapted the sentence:

*"Air parcels descending within an extratropical dry intrusion are expected to be drier and to create a stronger humidity gradient at the moisture source (increasing the d-excess) [...]",*

We have also added an analysis which confirms our hypothesis that the humidity gradient at the moisture source and not the local humidity gradient is decisive for the d-excess of the air parcels. This is shown in the Fig. R5, which also has been added to the manuscript (currently as Fig. 8). We have added the following sentences to Sect. 4 to discuss this new figure:

*"This interplay between large-scale subsidence, humidity gradient and d-excess is visualised in Fig. 8. The air parcels' d-excess at arrival is clearly more strongly influenced by the humidity gradient at the moisture source (Fig. 8a) than by the humidity gradient at their arrival location (Fig. 8b)."*

[Figure]

**Figure R5:** Relation between relative humidity with respect to sea surface temperature ($RH_{SST}$) and d-excess in vapour of the $COSMO_{iso,10km}$ air parcels at their arrival near cloud base. **(a)** $RH_{SST}$ at the moisture source calculated with the moisture source diagnostic by Sodemann et al., 2008) of the air parcels (Pearson correlation coefficient$= -0.92$; two-tailed p-value $= 1.2 \times 10^{-221}$); **(b)** $RH_{SST}$ at the air parcel's arrival location (Pearson correlation coefficient$= -0.37$; two-tailed p-value $= 3.2 \times 10^{-19}$). The data points are coloured according to the mean subsidence rate of air parcels during the last 6 d before arrival at cloud base ($\overline{\Delta z_{6d}}$, i.e., the variable shown on the y-axis of Fig. 7c). Shown are the mean values over the 138 $COSMO_{iso,10km}$ air parcels arriving simultaneously (from 22 January to 13 February 2020) at the three cloud base trajectory arrival levels (940, 920, and 900 hPa) in the domain 54.5-61° W and 11-16° N. The dashed black lines indicate the mean values of the respective $x$- and $y$-variable over all arrival time steps.

C.2 L181: "influenced by the large-scale circulation": same question as above: is it really the large-scale circulation that directly impact d-excess? Or is it the local surface relative humidity, impacted by the large-scale circulation, that impacts d-excess?

Please also refer to the answer to previous comment. We adapted this sentence to:

*"influenced by the large-scale circulation through the conditions created at the moisture source."*

**Minor comments:**

1.1 L6: what is the subject of "show"? Cut this sentence into 2 sentences for clarity?

We agree with the reviewer that this sentence was too long. We rewrote it as follows:

*"The three main findings are that (1) the contrasting isotope and humidity characteristics in clear-sky versus cloudy cloud base environments emerge due to vertical transport on time scales of several hours associated with local, convective circulations; (2) the isotope signals in these cloud base environments show a clear diel cycle shaped by shallow mesoscale cloud-relative overturning circulations;*

*and (3) the cloud base isotopes are, in addition, sensitive to variations in the large-scale circulation on time scales of several days, which shows on average a Hadley-type subsidence but occasionally much stronger descent related to extratropical dry intrusions.".*

1.2 L126-127: this technical comment could go in the Methods section, in 2.4.
Agreed, we shifted this sentence to Section 2.4 in the Methods.

1.3 Fig 5: I find it hard to see the relative proportions of shallow, middle and deep clouds. They all look flat. Is it possible to rather show the proportion of deep cloud as a line, like for the cloud fraction?
We revised Fig. 5 and split it into multiple figures to separate the diel cycles of cloud fraction at different altitudes, precipitation, and updraft from the diel cycles in isotope signals. The different cloud categories are no longer needed, with the new figure which shows cloud fraction as a function of height and informs more reliably about vertical growth and extent of clouds.

1.4 L136: why are the variations in cloudy patches small? Is it because the vapor comes from the near-surface, where the temporal variations are small?
Yes, exactly. The BCO mixed layer $q$ and $\delta^2$H are in phase and of about the same amplitude as the cloudy cloud base signals. We clarified this in the text by adding the following sentence:
*"The cloudy patches are fed by updrafts bringing moisture from the subcloud layer, in which the amplitude of the variability in $q$ and $\delta^2 H$ is small and of about the same extent as observed in clouds (see part 1 of this study and Appendix A)."*

1.5 L146: "subsidence rate": is it really a rate, or do you mean the altitude of origin? Same question l 200: "subsidence ... stronger": is the subsidence stronger or originating from a higher altitude? Or is it equivalent? (and if so, what process links the subsidence velocity with the origin altitude of air parcels?)
We mean the slantwise 12 h subsidence rate as determined by our trajectory approach. A larger slantwise subsidence rate is equivalent with air originating from higher altitude (since we consider the subsidence over a fixed time window of 12 h). Our approach with the 12 h subsidence seems to have caused some confusion. Therefore, we will adapt our approach by looking at the altitude of the air parcels when they were last inside a cloud instead of the subsidence rate. With the new approach, we get a direct estimate of the altitude of origin. This will simplify the discussion of the results in Sect. 3. We will adapt figures and the text accordingly.

1.6 L167: we don't know for d-excess: is it possible to extend the duration of the trajectories? I would make 2 separate sentences for $\delta^2$H and d-excess, since for $\delta^2$H the maximum link is for 4 days, whereas for d-excess it is for 6 days or larger, we don't know.
The trajectories reach the domain limits and can therefore not be extended within the same simulation domain. We adapted the text as suggested.

1.7 L170: "not" → "less" Done.

1.8 Table 1: is it possible to give some p-values associated with the correlation coefficients, or some correlation coefficient threshold to reach a given p-value?
Done. See new table in paper appendix and revised table in paper.

1.9 L184: this is interesting. Can you also elaborate on what are the mechanisms that overwrite q faster than $\delta^2$H?
We will add one or two sentences elaborating on these mechanisms.

1.10 L218-219: This idea is very interesting. This recalls me the study of [Brient et al., 2019] who identifies coherent structures in shallow clouds. It would be very interesting to look at the isotopic signature of these different sructures. Maybe this study could be cited.
Thank you, very interesting study indeed, we added the reference in the text.

1.11 Fig 8: it would be useful to add a Rayleigh curve and a mixing curve to guide the interpretation of these plots.
We will add these curves to the plot.

**Reviewer 2:**

**General comments**:

A I think the paper could benefit from a more detailed discussion in particular of section 3 when it comes to the cloud-scale circulations, where some aspects of the proposed mechanisms remain a bit unclear to me. A couple of the more detailed comments below relate to this point.
We will complement and significantly extended the discussion of section 3 with more details on the proposed mechanism (see also our response to reviewer 1, major point B).

B Knowing that the authors collected an extensive isotope dataset in the region and time frame that is considered here, I cannot help but wonder how the simulations, e.g., in terms of the diurnal cycle, compare to observed isotope signatures. Some of that might be covered in part 1 (not sure about the diurnal cycle though) but if would be great to see some of these observation (or conclusions from part 1) to be picked up here again.
The evaluation of the diel cycle of isotope signals at cloud base is very difficult, despite the extensive observational effort conducted during EUREC⁴A. But, we follow the reviewers' recommendation and will add an evaluation of the diel cycle of isotope observations from the near-surface mixed layer obtained from the Barbados Cloud Observatory in the Appendix of the paper and discuss it at the beginning of Sect. 3. See our answer to reviewer 1, major point A (Fig. R1-3) for more details.

**Detailed comments:**

2.1 Introduction: To make the paper more accessible to readers that are not familiar with isotopes (like me) it would be helpful to be more specific or basic in how one expects the different isotope measures to reflect atmospheric processes. E.g., is it correct that heavy isotopes are less likely to evaporate from the ocean and hence that atmospheric vapour is lighter than ocean water, i.e., $\delta^2$H is negative? Are heavy isotopes more likely to condense on cloud droplets so that condensation in clouds leaves even lighter (more negative $\delta^2$H) vapor behind? Which physical processes affect the relative variation of $\delta^2$H and $\delta^{18}$O? (Is an O-heavy isotope even more unlikely to evaporate than an H-heavy isotope?) In which way are d-excess values or tendencies affected by thermodynamic conditions? How does vertical transport (physically) affect $\delta^2$H or d-excess? Maybe it could be helpful add some of the explanation in the caption of Figure 1 to the main text.
We will add some additional background information about isotopes and which atmospheric processes influence their concentration in the introduction.

2.2 Figure 1: This is a nice illustration. Is it possible to indicate expected dH2 and d-excess values or tendencies in the figure, e.g., by colouring the background according to values or by adding tendencies to the arrows?
We will add this information in some way to the figure.

2.3 Figure 1 caption: "can have a depleting or enriching effect" On what does this depend?
There are several factors that need to be taken into account to assess whether below-cloud processes lead to an enrichment or a depletion of the vapour. The partial evaporation of rain droplets will deplete the surrounding vapour due to fractionation (i.e., heavy isotopes preferentially staying in the liquid phase and light isotopes more readily going into the vapour phase). The full evaporation of rain droplets will in most cases enrich the surrounding vapour, since all, heavy and light, isotopes go from the liquid into the vapour phase. However, if the rain droplets formed from much more depleted vapour (i.e., at much higher altitudes) than the surrounding vapour, their full evaporation might also lead to a depletion of the surrounding vapour. Thus, how strong the enrichment/depletion of the surrounding vapour is strongly depends on the original isotopic composition of the rain droplets. These explanations are too long to be added to the caption of Fig. 1. Instead, we now refer to two references (Aemisegger et al., 2015; Graf et al., 2019) where this background about below-cloud processes can be looked up.

2.4 It would be helpful to add a short explanation on why you focus on cloud base properties.
The interest in cloud base comes from the fact that climate models suggest that the cloud base cloud fraction of shallow trade-wind cumuli is very sensitive to changes in environmental conditions, while process models suggest the opposite (Bony et al., 2017, 2022). We have added the following

sentence to the introduction:

*"Especially, the cloud fraction at cloud base has been identified as a key parameter for the spread of the modelled feedback of these clouds to climate change (Bony et al., 2017). To shed light on the mechanisms controlling cloud base cloud fraction in the trades, [...]"*

2.5 Section 2.1: Although the simulations are described in Part 1, I would wish for a bit more details here. What are the different domains? Are there limitation due to shallow convection not being resolved at 1 - 10 km grid spacing? Where are trajectories started and which time step do they apply? Can you be more specific in how many trajectories you compute?

**Simulations:** We have added some more details about the simulations (resolution, temporal coverage, nudging) and the convection-resolving model setup in Sect. 2.1.

**Trajectories:**

- Note that details about the trajectories calculated with $COSMO_{iso,5km}$ and $COSMO_{iso,10km}$ are given in the respective section (Sect. 2.4 and 2.5). We now inform about this in Sect. 2.1 in the following way:

   *"$COSMO_{iso,1km}$ and $COSMO_{iso,5km}$ are used to characterise the cloud-relative circulation (Sect. 3), and $COSMOiso,10km$ to assess the large-scale circulation (Sect. 4). For this, air parcel backward trajectories are calculated with data from $COSMO_{iso,5km}$ and $COSMO_{iso,10km}$ using the Lagrangian analysis tool LAGRANTO (Wernli and Davies, 1997; Sprenger and Wernli, 2015, see detailed description of trajectory starting points in Sect. 2.4 and Sect. 2.5)."*

- **$COSMO_{iso,5km}$:** We start the backward trajectories at every hourly time step from all *dry-warm* cloud base grid points (red in Fig. 3d) and (as a response to reviewer comment 2.7) at hourly time steps from 1000 randomly selected *clear* cloud base grid points. Thus, the number of *dry-warm* backward trajectories varies from time step to time step. We now show the number in the revised Fig. 2 in terms of fraction (left y-axis) and absolute values (right y-axis). We now also clarify in the caption of Fig. 2 that the *dry-warm* cloud base grid points serve as starting point for backward trajectories. Furthermore, we adapted the text in Sect. 2.4 to the following for clarification (note that the *clear* backward trajectories are new as a response to reviewer comment 2.7; Fig. 3 was revised accordingly):

   *"To investigate the formation mechanism of the mesoscale features that form the dry-warm patches at cloud base, we calculate 24 h-backward trajectories. We start them every hourly timestep between 22 January and 13 February 2020 from all cloud base grid points identified as dry-warm in the domain 54.5-61° W and 11-16° N in the $COSMO_{iso,5km}$ simulation. Note that the number of dry-warm cloud base grid points varies from timestep to timestep (Fig. 2). Summed over all timesteps, this results in a total of 568'124 dry-warm trajectories. Similarly, we start 24 h-backward trajectories every hourly timestep between 22 January and 13 February 2020 from 1000 randomly selected cloud base grid points identified as clear in the $COSMO_{iso,5km}$ simulation. We fix the number of clear trajectories for computational reasons, since about 9'000 grid points are identified as clear every hourly timestep (not shown). Summed over all timesteps, this results in a total of 539'460 clear trajectories. The starting points of the dry-warm (red areas) and clear (yellow dots) backward trajectories are shown for an exemplary timestep in Fig. 3d."*

   - **$COSMO_{iso,10km}$:** In our opinion, the calculation of the $COSMO_{iso,10km}$ backward trajectories is sufficiently and clearly described in Sect. 2.5. We encourage the reviewer to specify what is unclear, if she disagrees with our opinion.

2.6 L68: "Every vertical profile": do you mean vertical profiles for each grid point?

Yes, each grid point in the considered domain. We changed the sentence to:

*"Vertical profiles at every grid point in the domain..."*

2.7 L92: Why are dry-warm points of particular interest?

We have added the following text (which also addresses reviewer comment 2.13) to clarify our interest in dry-warm points at the end of Sect. 2.3:

*"The reasoning to separate dry-warm and clear cloud base grid points stems from part 1 of this study. We assume that the characteristics of the dry-warm category result from coherent mesoscale subsidence and, therefore, can give insight into the downward branch of the overturning cloud-relative*

*circulation (sketched in Fig. 1). A dry-warm anomaly is expected below the downward branch be-
cause (1) subsidence causes adiabatic warming (generating a warm anomaly at cloud base), and
(2) a coherent mesoscale feature ensures a certain distance from clouds and through this minimises
mixing with moist air from surrounding clouds and avoids the influence of evaporating cloud and
rain droplets, thereby generating a dry anomaly at cloud base. Crucially, the absence of mixing
and phase changes throughout the subsidence leads to a conservation of the isotope signal within the
vapour from the point at which it was detrained from the cloud. As shown in the exemplary timestep
in Fig. 3, our definition of dry-warm indeed applies to grid points that are well away from clouds
and therefore are optimal to analyse the processes linked to mesoscale subsidence alone, which we
expect to be resolved in the simulations used here.*

*For the clear category, we assume that several processes (subsidence, turbulent mixing, evaporation
of cloud and rain droplets) lead to its characteristics. Some of these, e.g., turbulent mixing, happen-
ing on shorter temporal and spatial scales than resolved by our simulation output. Whether the two
cloud base environments, clear and dry-warm, truly emerge due to different processes is assessed
with backward trajectories, described in the next section.*

*Although we have a special interest in the dry-warm category, it is important to remember that
clear cloud base grid points cover a much larger area. Namely, about 81 % of the cloud base grid
points in COSMO$_{iso,1km}$ are categorised as clear and only about 8 % as dry-warm, considered over
the whole simulated period. This means, for instance, that for the mass balance at cloud base, the
clear category plays a much important role than the dry-warm one (knowing that downward winds
of similar magnitude prevail in both; see Villiger et al., 2023, their Fig. 15d)."*

To complement the analysis of dry-warm cloud base environments, we have added some additional
analysis of the clear cloud base environments. Namely (as already mentioned in the newly added
text above), we have calculated COSMO$_{iso,5km}$ backward trajectories from 1000 randomly selected
clear cloud base grid points at every time step. We do so to investigate the different processes
leading to the *dry-warm* and *clear* cloud base environments. Furthermore, we also analyse and
discuss the conditions in the clear cloud base environments. This has led to the following changes
in the manuscript:

- Fig. 3d: Adding of the 1000 randomly selected clear cloud base grid points used to calculate
  backward trajectories for the illustrative time step shown in the figure. Additionally, we no
  longer show cloud liquid water in panel (a) and (d), but the grid points identified as *cloud* to
  clearly show the three different cloud base categories *cloud*, *dry-warm*, and *clear*.

- Sect. 2.4: Adding of the description of the calculation of these 'clear' backward trajectories.

- New Fig. 6 (former Fig. 5) and associated text: Adding of diel cycle of specific humidity and
  $\delta^2$H in clear cloud base environments and discussion of these new results in the text.

- New figure and text discussing the 'clear' COSMO$_{iso,5km}$ trajectories.

2.8 L94: What is the reasoning to calculate hour-of-the-day deviation from the whole simulation period
instead of using day-specific hourly means?

Note that our data has an hourly temporal resolution, which means that the only option to calculate
day-specific hourly means is to calculate a mean over space, e.g., over the considered domain (*rel. to
domain mean* in Fig. R6 and R7). If we do so, we end up with dry-warm cloud base grid points that
reflect the spatial East-West gradient of the cloud base temperature (Fig. R6c), i.e., all dry-warm
grid points in the warmer western part of the domain (Fig. R6c).

Here, we are not interested in the East-West gradient of cloud base temperature, but in the mesoscale
subsidence (leading to the dry-warm cloud base grid points; see also answer to reviewer comment
2.7) occurring around clouds everywhere in the considered domain. Therefore, we decided to define
the dry-warm characteristics relative to grid-points wise mean values. We can define grid-point wise
mean values (1) over the whole period (e.g., one mean value per grid point; *rel. to overall mean*
in Fig. R6 and R7) or (2) for each hour-of-the-day over the whole period (e.g., 24 mean values per
grid point; *rel. to hour-of-day mean* in Fig. R6 and R7).

Using the definition *rel. to overall mean* yields slightly more *dry-warm* grid points (8.6 % over
the whole period) than using the definition *rel. to hour-of-day mean* (8 % over the whole period).
Thus, the *rel. to hour-of-day mean* definition might be seen as the more restrictive one. The spatial
(Fig. R6a,b) as well as the temporal (Fig. R7) distribution of the two differently identified *dry-warm*
cloud base grid points is very similar, leading to the conclusion that the distinction between the

two definitions is not decisive for the key findings of our study. Based on this and for consistency with part 1 of this study (Villiger et al., 2023), we have not adapted the definition applied in the paper.

[Figure]

**Figure R6:** Spatial distribution of **(a-c)** cloud base grid points identified as *dry-warm*, **(d)** cloud liquid water (QC), **(e)** specific humidity (QV), and **(f)** potential temperature ($\theta$) at cloud base at 15 UTC on 2 February 2020. *Dry-warm* cloud base grid points are identified as deviations in specific humidity and potential temperature (see paper for details) relative to **(a)** the domain mean of the current time step, **(b)** the grid-point-wise mean over the whole period, or **(c)** the grid-points-wise hour-of-day mean over the whole period. The fraction of grid points in the domain identified as *dry-warm* is **(a)** 4.3 %, **(b)** 6.3 %, and **(c)** 9.8 %. Shown is the data from COSMO$_{iso,5km}$ in the domain 54.5-61° W and 11-16° N and the track flown by the research aircraft ATR (cf. Villiger et al., 2023) in pink for scale. Note that similar results are obtained with the data from COSMO$_{iso,1km}$. Data: COSMO$_{iso,5km}$.

[Figure]

**Figure R7:** Hourly time series of the fraction (left axis) and number (right axis) of cloud base grid points identified as *dry-warm* in the domain 54.5-61° W and 11-16° N for COSMO$_{iso,5km}$. Three different definitions of *dry-warm* are applied (see response to reviewer comment 2.8). Data: COSMO$_{iso,5km}$.

2.9 L105: How do you choose starting points? Or do you start from all dry-warm cloud base grid points?

See answer to reviewer comment 2.5.

2.10 The naming convention of variables in Fig. 4 is not clear to me.

We will adapt the naming convention of the variables in Fig. 4 and reformulated the figure caption for more clarity.

2.11 L126: How do vertical velocities between the 1-km and the 5-km compare? Do cloudy updraft strengths differ?

Cloudy updraft (i.e., named *mass flux* in Fig. R8) strengths are stronger in $COSMO_{iso,1km}$ than in $COSMO_{iso,5km}$ (Fig. R8a). However, most importantly, the diel cycles are similar in the two datasets (Fig. R8b). The same is true for the negative velocities in *dry-warm* cloud base grid points (named *negative mass flux* in Fig. R8; which might be seen as the downward branch of the cloud-relative overturning circulation, see also reviewer comment 2.7): Downward velocities in *dry-warm* cloud base grid points are stronger in the $COSMO_{iso,1km}$ data than in the $COSMO_{iso,5km}$ data (Fig. R8c), but the diel cycles are very similar (Fig. R8d). We do not think that the differences of updraft strengths between the $COSMO_{iso,5km}$ and $COSMO_{iso,1km}$ are relevant for our findings, because we are not primarily interested in their magnitude but rather in their variation over the course of the day and the impact of this variation on cloud base isotopes (i.e., the coupling between the variables). We will clarify this at the beginning of Sect. 3.

2.12 L137: "largely unaffected": Please explain how you arrive at this interpretation. Because the signal is small?

Yes, exactly. The BCO mixed layer $q$ and $\delta^2 H$ are in phase and of about the same amplitude as the cloudy cloud base signals. We clarified this in the text by adding the following sentence:

*"The cloudy patches are fed by updrafts bringing moisture from the subcloud layer, in which the amplitude of the variability in $q$ and $\delta^2 H$ is small and of about the same extent as observed in clouds (see part 1 of this study and Appendix A)."*

2.13 L140: Can you check how well it is closed by multiplying area fractions and vertical velocities of both branches?

From part 1 of this study (Table 4 and Fig. 15), we know the fraction of cloud base grid points categorised as *dry-warm*, *clear*, (non-precipitating) *cloud*, and (precipitating) *cloud-rain* as well as the mean vertical velocity for each of these categories. We can use these numbers for a first rough mass balance analysis (Table R2). The mass fluxes at cloud base are not fully balanced in $COSMO_{iso,1km}$ considered over the domain 54.5-61° W and 11-16° N and over the period 20 January to 13 February 2020. Adding the estimated mass fluxes from the different cloud base categories (Table R2), we get a negative term of $-387'637 \times 10^6 \ m^3 \ s^{-1}$. Note that this is a very rough estimate, since we summarized mass fluxes over a considerably large domain and time period. However, the key message of the analysis is that the upward mass flux of clouds (*cloud* and *cloud-rain*) is fully balanced by the downward mass flux occurring in the *clear* cloud base grid points. The contribution of the *dry-warm* cloud base grid points to the negative mass flux is negligible in comparison. The dominance of the *clear* category over the *dry-warm* category in terms of negative mass flux comes from its much larger areal coverage.

Considering the importance of the *clear* cloud base category for the mass balance, we will add it to our analysis of the diel cycle in Sect. 3. Please also refer to our response to reviewer comment 2.7.

2.14 L144: It is not clear to me why "the amount of vapour that returns to the cloud base is less"? Because it is colder higher up and therefore saturated q is less?

Yes. We will clarify this in the text.

2.15 149-153: This is not clear to me. Why does the min-max time shift in dry-warm patch characteristics determine the point in time when subsiding air detrained from the cloud?

Please also refer to our response to reviewer 2 major point B. With our new approach of looking for each trajectory's last time in a cloud (i.e., point of last saturation) instead of the subsidence rate over 12 h, we aim at clarifying this point of confusion. We will adapt the text accordingly.

2.16 L154-159: How do your results compare to literature, e.g. observed circulations from George et al. or the diurnal cycle from Vial et al.?

[Figure]

**Figure R8:** **(a,b)** Mass flux (mean vertical velocity of all cloudy cloud base grid points with positive vertical velocities) **(c,d)** "negative mass flux" (mean vertical velocity of all dry-warm cloud base grid points with negative vertical velocities; only time steps with at least 100 grid points are considered) over whole simulated period for $COSMO_{iso,1km}$ (red) and $COSMO_{iso,5km}$ (red). Shown are **(a,c)** hourly values and **(b,d)** the diel cycle (25-75 percentile range as shading and the median as line). Calculated with data from domain 54.5-61° W and 11-16° N during the period 20 January to 13 February 2020 are shown.

We will add some sentences to Sect. 3, in which we compare our results to the ones from Vial et al. (2019), Vogel et al. (2020,2022), and George et al. (2023).

2.17 Fig. 6 and 8 are barely discussed in the text.
We will extend the discussion of these figures in the text.

2.18 L200: Why does a deeper circulation need to be stronger?
Because we looked at it over a constant period of time (i.e., 12h). A stronger subsidence over 12 h implies that the air parcels originate from higher altitude. We will ensure that our reasoning is more clearly stated in Sect. 5.

2.19 L204: Do you refer to the few points with low d-excess around q = 8-9 g/kg here? They don't seem special to me in terms of large-scale subsidence (Fig. 8b).
No, we refer to the points with strong large-scale subsidence (most of them from 22 January 2020;

**Table R2:** Fraction and number of cloud base grid points identified as *dry-warm*, *clear*, *cloud*, and *cloud-rain* in the COSMO$_{iso,1km}$ simulation in the domain 54.5-61° W and 11-16° N during the period 20 January to 13 February 2020. For each cloud base category, the mean vertical velocity is calculated and the mass flux (as the area corresponding to the number of 1×1 km grid points multiplied by the mean vertical velocity. Note that we distinguish between precipitating clouds (*cloud-rain*) and non-precipitating clouds (*cloud*) because the analysis shown here is taken from part 1 of this study (cf. Table 4 and Fig. 15 in Villiger et al., 2023).

| Cloud base category | Fraction of grid points [%] | Number of grid points | Mean vertical velocity [m s$^{-1}$] | Mass flux [×10$^6$ m$^3$ s$^{-1}$] |
|---|---|---|---|---|
| *dry-warm* | 7.6 | 14'262'075 | -0.01 | −142'620.75 |
| *clear* | 81.2 | 152'943'787 | -0.02 | −3'058'875.74 |
| *cloud* | 10.9 | 20'597'884 | 0.12 | 2'471'746.08 |
| *cloud-rain* | 0.3 | 645'497 | 0.53 | 342'113.41 |

encircled black dashed in Fig. R9). We will clarify this by encircling them in the figure and refer to them in the text correspondingly.

[Figure]

**Figure R9:** Same as Fig. 8b of the paper, but with the data points of interest highlighted by encircling them (black dashed).

**Technical comments**:
Please check figures, their captions and references in the text carefully.
We will check this thoroughly.

I noted a few typos/corrections below as technical comments.

2.1 L76: "761, 914, and 1082m" this is not clear
   We changed this sentence to *"The hourly cloud base levels alternate between 783 and 970 m for COSMO$_{iso,10km}$ and change between three levels, i.e. 761, 914, and 1082 m, for COSMO$_{iso,5km}$ and COSMO$_{iso,1km}$."*

2.2 L107: There is no Fig 4b Done.

2.3 Fig. 2: caption and axis labels do not match This conflict has been resolved.

2.4 Fig. 3: add "(red)" after "dry-warm"; (b) → (d) Done, both as suggested.

2.5 Fig. 5: red and green as contrasting colours are not optimal for colour-blind readers; in c please use dashed line style also to distinguish the thin lines
   In Fig. 6 we changed the Figure as suggested and used blue instead of green for the clouds. The isotope signals are also clearly stratified along the vertical axis such that a confusion of the groups can be avoided.

2.6 Fig. 7: What do the two different dates per plot mean?
The black date on the left of the panels show the arriving times, the grey dates 3.5 days before arrival. We clarified this in the caption.

2.7 L135: delete "d-e" Done.

**References**

[revised manuscript text omitted]

---

## Author Response (AR1)

**Authors' response to reviews of the manuscript**

**"Water isotopic characterisation of the cloud-circulation coupling in the North Atlantic trades. Part 2: The imprint of the atmospheric circulation at different scales"**

by Leonie Villiger and Franziska Aemisegger

We thank the anonymous reviewer (hereafter reviewer 1) and Ann Kristin Naumann (hereafter reviewer 2) for their insightful comments, which we address in detail in our responses below. They helped us to improve the presentation of our results and led to the following main changes in the revised manuscript:

1. We included a short evaluation of the diel cycles in vapour isotopes in the appendix using observations from a near-surface coastal site as a response to **reviewer 1 major point A and reviewer 2 major point B**.

2. We added more background information about isotopes in the introduction, as well as more details about the COSMO$_{iso}$ simulations in Sect. 2.1 (Datasets) as a response to **reviewer 2 major point A and comment 2.5**.

3. We extended the discussion and improve the presentation of the diel cycle in Sect. 3 (Cloud base isotopes and the cloud-relative circulation) as a response to **reviewer 1 major point B and reviewer 2 comments 2.7, 2.13, and 2.15**. These changes also led to adaptations of the text in Sect. 2 (Data and methods) and Sect. 5 (Summary and conclusion).

4. We added a short analysis in Sect. 4 (Cloud base isotopes and the large-scale circulation) showing that the d-excess of vapour is more strongly influenced by the humidity gradient at the moisture source than by the humidity gradient at a given point of observation as a response to **reviewer 1 major point C**.

In addition we revised the abstract to align it more closely with the ACP guidelines (see also response to reviewer comment 1.1).

Please find our detailed response to the reviewers' comments below in the following format:
Reviewer's comment. Authors' response.
* * *
**Reviewer 1:**

**General comment**: The article is well written and well illustrated. However, while I found part 1 too long, I found this paper too short.
We thank the reviewer very much for her/his constructive and insightful comments, which have helped us to improve the clarity of the manuscript. The first submitted version of this paper was very short because it was initially planned as a letter. We adapted the manuscript providing more detail in the text, where the reviewers asked for more discussion. We also implemented additional analyses (in particular on the diel cycle), which aim at facilitating the understanding of the manuscript.

**Major comments**:

A **What is the realism of the simulated diel cycle of cloud fraction, cloud depth and water isotopes, with respect to observations?** I don't remember that this was evaluated in part 1. In particular, the amplitude of the diel cycles and the relative phasing for different variables were not evaluated.
In part 1, we did evaluate many aspects of isotope signals associated with shallow cumulus cloud formation and their environment, but not the daily cycle because observations of the daily cycle at cloud base are not available and because we wanted to treat this aspect in part 2.

A.1 Is there enough observations to evaluate the simulated diel cycle?

Here we split our response in a discussion of non-isotope and isotope variables.

**Non-isotope variables**: The diel cycle of cloud fraction, cloud depth, precipitation, and convective mass flux at cloud base (to some extent comparable to the updraft strength considered in Fig. 5) has been evaluated in detail in previous publications and the characteristics of the diel cycle found here in COSMO$_{iso}$ are very close to the findings in Vial et al. (2019) and Vogel et al. (2020,2022), in which Large-Eddy simulations with ICON have been combined with observations. We added references to these studies in the manuscript as described in the response to the reviewer comment 2.16.

**Isotope variables**: Robust observations of the diel cycle of isotope signals at cloud base are very difficult to obtain. We have aircraft observations from different times over the day but despite the considerable deployment effort, we do not have enough data to cover the whole diel cycle at the cloud base level and for the different cloud base features in a robust way (see Fig. R1). In qualitative terms, the observations we have from the ATR point towards a minimum in $\delta^2$H in the morning hours, which is delayed compared to the minimum in $q$. But this is very speculative given the 18 available four-hour flights spread between 3 and 18 LT.

We do have ground-based observations from the Barbados Cloud Observatory (BCO), which we show in Fig. R2 together with the lowest model level data from the three COSMO$_{iso}$ simulations using the closest grid points (see Fig. R3). The diel cycles in isotope signals near the surface and at cloud base (between 780 and 970 m a.s.l.) do not necessarily have to be synchronous or of the same amplitude, because different processes may dominate at the two levels:

- stronger and more immediate influence of surface evaporation for the near-surface BCO data
- potential impact of land surface processes for the BCO site compared to the free ocean location at which the daily cycle at cloud base is analysed.

When comparing the simulation data in Fig. R2a,b (diel cycle of near-surface BCO $q$ and $\delta^2$H) to the cloudy group in the new Fig. 6 (former Fig. 5) of the manuscript (diel cycle of cloud base $q$ and $\delta^2$H in clouds), we observe very similar timings of minima (3-6 LT) and maxima (15-21 LT) in $q$ and $\delta^2$H. This is plausible, since the clouds are feed primarily by humidity from the mixed layer and thus near-surface isotope and humidity signals are most likely to match with cloudy signals at cloud base. We observe a delayed diel cycle at cloud base compared to the near-surface BCO site, when comparing the BCO data to the clear-sky and dry-warm patches: the maximum at BCO occurs earlier in the day (12-21 LT) in all three simulations, with a double peak in the 10 km simulation (15 LT and 2 LT). This delay is most likely due to the turnover time of the shallow circulation. Surface evaporation more strongly and more immediately impacts the near-surface BCO site. Furthermore, mixed layer isotope signals are affected by below cloud interaction with the falling rain. The amplitude of the diel cycle at cloud base is slightly larger for both $q$ and $\delta^2$H compared to the BCO, most likely owing to the stronger direct influence of dry and depleted air subsiding from cloud top regions and the much weaker impact of turbulent upward mixing of moisture from fresh ocean evaporation.

For the BCO, we see a relatively good agreement between the observed and the simulated diel cycles, both in terms of amplitude (Table R1) and phasing (Fig. R2). Shortcomings in the model are linked to (1) a too small amplitude of the diel cycle in the simulated $d$-excess (in agreement with the underestimation of the observed variability in $d$-excess by the COSMO$_{iso}$ simulations discussed in part 1) and (2) a slightly too late increase in $\delta^2$H in the morning hours. From this short comparison with the observational BCO data, we conclude that the physical ingredients shaping the diel cycle are adequately represented in the model.

We would like to emphasise here, that despite the fact that we cannot evaluate the diel cycle of isotopes in the model in detail with direct observations, we have evaluated many variables connected to it in a physical way (such as cloud base cloud fraction, precipitation isotope signals at cloud base and in the mixed layer; see Villiger, 2022).

We included Fig. R2 and Fig. R3 in the appendix of the manuscript and added the following note about this short evaluation of the daily cycle using the BCO observations in the manuscript at the beginning of Sect. 3:

*"Before we move on to exploring the potential coupling between the cloud-relative circulation and cloud base isotope signals, we have to briefly address the evaluation of the diel cycles with observations. The modelled diel cycles cannot be evaluated directly with observations due to the*

*lack of available observations at cloud base in different environments (cloud, clear, dry-warm) over the entire day. Instead, an evaluation of the diel cycle from observations at a close-by land site along the east coast of Barbados (BCO site, see Bailey et al. 2023) has been done, which shows a good agreement with the model (Appendix A). The diel cycles of $q$ and $\delta^2 H$ in clouds at cloud base are in phase with and of the same amplitude as the respective variable at the coastal site. A slight delay in phasing of the diel cycles in the clear and dry-warm patches at cloud base compared to the near-surface coastal point is due on the one hand to the circulation including the condensational depletion in clouds (see also discussions below), and, on the other hand, to the stronger direct influence of surface evaporation as well as below cloud interaction with falling rain at the BCO. From this short comparison with the observational BCO data, we conclude that the physical ingredients shaping the diel cycle are adequately represented in the model."*

[Figure]

**Figure R1:** Diel cycles of cloud base $\delta^2 H$ (left axis) and specific humidity (right axis) from $COSMO_{iso,1km}$ in different cloud base environments (as in Fig. 5 [now 6] of the paper). Median (marker) and the 25-75-percentile range (transparent line/errorbars) of cloud base $\delta^2 H$ (left axis) and specific humidity (right axis) for each ATR flight with isotope observations (RF03-RF20). Refer to Bailey et al. (2023) for a description of the ATR isotope observations.

[Figure]

**Figure R2:** Diel cycles of near-surface (a) specific humidity, (b) $\delta^2$H, and (c) d-excess in the COSMO$_{\text{iso,10km}}$ (teal), COSMO$_{\text{iso,tkm}}$ (yellow), and COSMO$_{\text{iso,1km}}$ (red) simulations together with observations collected during EUREC$^4$A at the Barbados Cloud Observatory (BCO; black, labelled BCO$_{\text{Picarro}}$); see Bailey et al. (2023) for a description of the BCO observations). For all four datasets, the period from 20 January to 13 February 2020 is considered for the calculation of the diel cycle. For COSMO$_{\text{iso}}$, the data from the lowest model level and the grid point closest to the BCO is selected (Fig. R3). Note that the lowest model level is different for the three COSMO$_{\text{iso}}$ setups and does not correspond to the altitude of the observations. Therefore, different y-axes are used for the simulations (left) and the observations (right).

[Figure]

**Figure R3:** Grid point of the COSMO$_{iso,10km}$ (teal), COSMO$_{iso,5km}$ (yellow), and COSMO$_{iso,1km}$ (red) model setup closest to the BCO (black marker). The altitude of the lowest model level is given in brackets (in meter above sea level [m a.s.l.]. The BCO is located 17 masl (Stevens et al., 2016, their Fig. 2) and the isotope observations were conducted at ∼ 3 m above ground (Bailey et al., 2023)

**Table R1:** Amplitude (maximum - minimum) of the diel cycles shown in Fig. R2 R2.

|  | COSMO$_{iso,\ 10km}$ | COSMO$_{iso,\ 5km}$ | COSMO$_{iso,\ 5km}$ | BCO$_{Picarro}$ |
|---|---|---|---|---|
| $q$ | 0.84 | 0.73 | 0.67 | 0.54 |
| $\delta^2$H | 1.23 | 1.32 | 1.08 | 1.84 |
| d-excess | 0.33 | 0.31 | 0.29 | 1.37 |

A.2 If not, how risky is it to base all the discussion on the role of mesoscale circulation on the diel cycle?

Radiative forcing and the resulting diurnal cycle is a first order control on the dynamics of shallow cumulus clouds. We therefore argue that by studying mesoscale circulations without taking the diel cycle into account, we would neglect a key driver of variability in the trade wind cloudiness. Furthermore, the focus on the diel cycle provides a useful and simple framework to study the physical mechanisms driving shallow convection. Finally, with the studies by Vial et al. (2019) and Vogel et al. (2020) there is already a solid literature basis on the diurnal cycle of shallow cumulus clouds in the western tropical North Atlantic, on which we can build our analysis that introduces isotopes as integral tracers of moist processes.

A.3 Isn't there a way to investigate the role of mesoscale circulations without relying on the diel cycle? e.g. by looking directly at dry-warm patches anomalies, independently of the diel cycle?

We do indeed have several ideas for new investigations on dry-warm patches in future research, but we argue that the radiative forcing driving the strength of mesoscale circulations is a first order control on isotope signals associated with trade wind cumulus cloud formation that should be investigated first.

B **Discussion in section 3 is too short.** I think Fig 5 (now split into two figures) leads to plenty of questions and I found it very stimulating, so I was quite frustrated at the end of section 3 that the discussion was so short and that I was left with so many unanswered questions.

We understand this frustration. This was due to our initial choice of keeping the manuscript in a very compact letter format. We extended the discussion of Sect. 3 (which also led to changes in Sect.2 and 5) and also added new analyses (also as a response to the reviewer comments 1.3, 2.7, 2.13, and 2.15). Below, we first answer your comments B.1-B.3 followed by a detailed description of the changes to the manuscript.

For example:

**B.1** Is the delay of the $\Delta z$ for 12h relative to 1h just the impact of the averaging back in time?

Yes, the delay of the $\Delta z$ for 12h relative to 1h is the impact of averaging over a longer time period backward in time. This effect is illustrated in Fig. R4, which shows the diel cycle of $\Delta z$ over different time windows. $\Delta z$ for 1 h, for example, only informs about the vertical displacement that the air parcels experienced during the last hour before they arrive at cloud base (i.e., provides an almost instantaneous description of the air parcels' subsidence). This 1 h-subsidence is strongest for the air parcels arriving at cloud base at approx. 4 LT. However, the 1 h-subsidence does not inform about the vertical displacement that the air parcels underwent earlier in their journey. Considering the vertical displacement over longer time periods (i.e., following the air parcels further back in time) gives a more robust estimate of the overall subsidence of the air parcels.

[Figure]

**Figure R4:** Diel cycle of $\overline{\Delta z}$ over different time windows (following the air parcel trajectories 1 to 24 h backward in time) derived from the $COSMO_{iso,5km}$ trajectories. Similar to Fig. 5b in the original version of the paper.

**B.2** Why is the $\delta^2H$ minimum in dry-warm patches delayed relative to the precipitation maximum and to the maximum subsidence? Is it because (1) $\delta^2H$ has some memory as it integrates processes back in time? Or is it because (2) it instantaneously adapts to some other variable that is delayed relative to the precipitation maximum, e.g. the cloud-base cloud fraction? In case of the (2), what would be the mechanism?

We strongly argue for (1): the isotope signal of the dry-warm patches is shaped primarily in the cloudy updrafts where condensation takes place. During subsequent subsidence in the dry-warm blobs, the isotope signal is conserved while being transported downward. The delay between the precipitation maximum and the $\delta^2H$ minimum in dry-warm patches is therefore due to the transport time from the updraft, where precipitation forms and, where the condensational depletion happens until the arrival of the vapour and its depleted isotope signal back at cloud base. The delay between the $\delta^2H$ minimum in dry-warm patches and the 1-h subsidence is due to the turnover time of the mesoscale shallow circulations. The $\delta^2H$ diel cycle is in phase with the 12h subsidence rates, indicating that the turnover times must be in this time range.

About process (2) suggested by the reviewer: The fact that the diel cycle of $\delta^2H$ in dry-warm patches is in phase with the cloud fraction is due to an indirect effect. Cloud fraction does not directly control $\delta^2H$ in dry warm patches. But cloud fraction is minimum when the subsidence strength integrated over the turnover times of the mesoscale circulations is maximum.

**B.3** Why is the cloud-base cloud fraction minimum delayed relative to the precipitation maximum? L129-130 is not convincing because cloud fraction does not look in phase with cloud depth.

This is due to the subsidence-delay discussed above. As a response to comment 1.3, we split the old Fig. 5 into Fig. 5 and 6. and now show the details of the cloud fraction at different heights. This new illustration of cloud fraction in Fig. 5 more clearly shows the link between cloud vertical depth, updraft strengths and precipitation. We reformulate the text in the following way: *"The reduction of clouds at low levels is associated with a progressive deepening of the clouds (Fig. 5a), which simultaneously leads to an increase in precipitation (Fig. 5b). The deepest clouds associated with the highest rain rates are observed around 06 LT (10 UTC)."*

Comments B.1 - B.3 (as well as comments 1.5 and 2.15) showed that our approach of looking at

$\Delta z$ over the last hours before the air parcels arrive at cloud base (derived from COSMO$_{\text{iso,5km}}$ trajectories) as a proxy for the strength of the cloud-relative circulation led to a lot of confusions. We, therefore, decided to replace this approach with a more straightforward one. Namely, we now determine the altitude of each air parcel when it was last inside a cloud (i.e., the cloud detrainment level) based on the cloud liquid water along the COSMO$_{\text{iso,5km}}$ trajectories. This replacement has led to the following changes:

- A description of how the cloud detrainment altitude is derived from COSMO$_{\text{iso,5km}}$ trajectories was added to Sect. 2.4 including an update of Fig. 4 displaying a schematic of the described procedure.

- Discussion of the novel results in Sect.3 and Sect.5, replacing the text of the earlier results based on the $\Delta z$ approach. In the same manner, Fig. 5 (now split into Fig. 5 and 6) and Fig. 8 (now Fig. 10) have been updated. Here, it is important to note, that the key findings of the analysis of the cloud-relative circulation and its imprint on cloud base isotopes do not change. The air parcels arriving in dry-warm cloud base patches are detrained from clouds 10-14 hours earlier (updated Fig. 5), which on average matches with the 12 h subsidence timescale identified with the $\Delta z$ approach. Moreover, the cloud detrainment altitude explains the degree of drying and depletion in the dry-warm patches, similarly as the $\Delta z$ over 12 h in the pervious version of the manuscript.

**C  Relative impact of local processes and large-scale circulation on d-excess?**

C.1  L177-178: "humidity gradient": where? locally, or at the moisture source? If dry intrusions are associated with a drier near-surface air locally and thus higher d-excess in the surface evaporation flux locally, we would expect a good correlations between d-excess and surface relative humidity. Is it the case?
We meant the humidity gradient at the moisture source. We adapted the sentence:
*"Air parcels descending within an extratropical dry intrusion are expected to be drier and to create a stronger near-surface humidity gradient (i.e., lower $RH_{\text{SST}}$) at the moisture source (increasing the d-excess) than air parcels crossing the North Atlantic at low levels within the comparably moist trade winds where frequent detrainment of cloudy air occurs (lowering the d-excess; Fig. 1b)."*,
We also added an analysis which confirms our hypothesis that the humidity gradient at the moisture source and not the local humidity gradient is decisive for the d-excess of the air parcels. This is shown in the new Fig. 8. We have added the following sentences to Sect. 4 to discuss this new figure:
*"This interplay between large-scale subsidence, humidity gradient and d-excess is visualised in Fig. 8. The air parcels' d-excess at arrival is clearly more strongly influenced by the humidity gradient at the moisture source (Fig. 8a) than by the humidity gradient at their arrival location (Fig. 8b)."*
Since we used an additional tool (the moisture source diagnostic by Sodemann et al., 2008) for this analysis, we added a description of it in Sect. 2.5 on L187-194. As Sect. 2.5 no longer only addresses the subsidence in the large-scale circulation, we renamed it to *"Lagrangian characterisation of the large-scale circulation"* (for consistency, we also renamed Sect. 2.4 to *"Lagrangian characterisation of the cloud-relative circulation"*).

C.2  L181: "influenced by the large-scale circulation": same question as above: is it really the large-scale circulation that directly impact d-excess? Or is it the local surface relative humidity, impacted by the large-scale circulation, that impacts d-excess?
Please also refer to the answer to previous comment. We adapted this sentence to:
*"[...] influenced by the large-scale circulation through the conditions created at the moisture source."*

**Minor comments:**

1.1  L6: what is the subject of "show"? Cut this sentence into 2 sentences for clarity?
We agree with the reviewer that this sentence was too long. Also, we noted that our abstract doesn't follow the ACP guidelines (`https://www.atmospheric-chemistry-and-physics.net/policies/guidelines_for_authors.html`). Therefore, we rewrote the abstract with the goal

to make it clearer and align it with the guidelines. The sentence, referred to in this comment, no longer exists in this form.

1.2 L126-127: this technical comment could go in the Methods section, in 2.4.
We moved the sentence *"We use COSMO$_{iso,5km}$ for these trajectories because COSMO$_{iso,1km}$ has too small a domain to trace air parcels over several hours."* to Sect. 2.4, L172-174."

1.3 Fig 5: I find it hard to see the relative proportions of shallow, middle and deep clouds. They all look flat. Is it possible to rather show the proportion of deep cloud as a line, like for the cloud fraction?
We revised Fig. 5 and split it into two figures (Fig. 5 and Fig. 6) to separate the diel cycles of cloud fraction at different altitudes, precipitation, updraft strengths and cloud detrainment altitude from the diel cycles in isotope signals. We believe the discussion in Sect. 3 is easier to follow if first the variables in Fig. 5 are addressed, and then the variables in Fig. 6 (see also response to reviewer 1's major comment B). The different cloud categories are no longer needed, since the new figure which shows cloud fraction as a function of height and informs more reliably about vertical growth and extent of clouds. The removal of the cloud categories led to the removal of their description from Sect. 2.3 and adaptations of the text in Sect. 3.

1.4 L136: why are the variations in cloudy patches small? Is it because the vapor comes from the near-surface, where the temporal variations are small?
Yes, exactly. The BCO mixed layer $q$ and $\delta^2H$ are in phase and of about the same amplitude as the cloudy cloud base signals. We clarified this in the text with the sentence:
*"The cloud patches are fed by updrafts bringing moisture from the subcloud layer, in which the amplitude of the variability in q and $\delta^2H$ is small and of about the same extent as observed in clouds (see part 1 of this study and Appendix A). While the cloudy grid points are thus largely unaffected by the diel cycle of convection, [...]"*

1.5 L146: "subsidence rate": is it really a rate, or do you mean the altitude of origin? Same question l 200: "subsidence ... stronger": is the subsidence stronger or originating from a higher altitude? Or is it equivalent? (and if so, what process links the subsidence velocity with the origin altitude of air parcels?)
We meant the slantwise 12 h subsidence rate as determined by our trajectory approach. A larger slantwise subsidence rate is equivalent with air originating from higher altitude (since we considered the subsidence over a fixed time window of 12 h). Our approach with the 12 h subsidence caused confusion, which is why we replaced it with the alternative approach described in the response to reviewer 1's major comment B. This replacement also led to the replacement of the sentence referred to in this comment.

1.6 L167: we don't know for d-excess: is it possible to extend the duration of the trajectories? I would make 2 separate sentences for $\delta^2H$ and d-excess, since for $\delta^2H$ the maximum link is for 4 days, whereas for d-excess it is for 6 days or larger, we don't know.
The trajectories reach the domain limits and can therefore not be extended within the same simulation domain. We adapted the text as suggested.

1.7 L170: "not" $\rightarrow$ "less" Done.

1.8 Table 1: is it possible to give some p-values associated with the correlation coefficients, or some correlation coefficient threshold to reach a given p-value?
Done. See new table in paper appendix and revised table in the paper.

1.9 L184: this is interesting. Can you also elaborate on what are the mechanisms that overwrite q faster than $\delta^2H$?
We added the following sentence to Sect. 4:
L317-321: "This difference in memory of the history of moist processes along the flow in $q$ and $\delta^2H$ is due to the fact that $q$ alone is determined to first order by the temperature just before the detrainment from clouds (i.e., last saturation paradigm, see Sherwood, 1996; Sherwood et al., 2010), while the $\delta^2H$ contains information on both $[^1H_2^{16}O]$ and $[^1H^2H^{16}O]$, which relates to the condensation history in the clouds. The d-excess in turn connects to the non-equilibrium conditions at the evaporative moisture source, while being at first order unaffected by equilibrium cloud processing."

1.10 L218-219: This idea is very interesting. This recalls me the study of [Brient et al., 2019] who identifies coherent structures in shallow clouds. It would be very interesting to look at the isotopic signature of these different sructures. Maybe this study could be cited.

*Thank you, very interesting study indeed, we added the reference in the text.*

1.11 Fig 8: it would be useful to add a Rayleigh curve and a mixing curve to guide the interpretation of these plots.

*We added the requested lines in the figure and discuss them on L387-389:*

*In Fig. 10 the $\delta^2 H$-q value pairs of dry-warm cloud base environments together with reference lines illustrating the impact of mixing (dark gray dashed lines) and precipitation production in clouds (assuming 100 % precipitation efficiency; light gray lines) show that a combination of processes is involved including mixing and microphysical processes.*
* * *
**Reviewer 2:**

**General comments**:

A I think the paper could benefit from a more detailed discussion in particular of section 3 when it comes to the cloud-scale circulations, where some aspects of the proposed mechanisms remain a bit unclear to me. A couple of the more detailed comments below relate to this point.

*We complemented and significantly extended the discussion of section 3 with more details on the proposed mechanism. See our response to reviewer 1, major point B for more details.*

B Knowing that the authors collected an extensive isotope dataset in the region and time frame that is considered here, I cannot help but wonder how the simulations, e.g., in terms of the diurnal cycle, compare to observed isotope signatures. Some of that might be covered in part 1 (not sure about the diurnal cycle though) but if would be great to see some of these observation (or conclusions from part 1) to be picked up here again.

*The evaluation of the diel cycle of isotope signals at cloud base is very difficult, despite the extensive observational effort conducted during EUREC$^4$A. But, we follow the reviewers' recommendation and added an evaluation of the diel cycle of isotope observations from the near-surface mixed layer obtained from the Barbados Cloud Observatory in the Appendix of the paper and discuss it at the beginning of Sect. 3. See our answer to reviewer 1, major point A (Fig. R1-3) for more details.*

**Detailed comments:**

2.1 Introduction: To make the paper more accessible to readers that are not familiar with isotopes (like me) it would be helpful to be more specific or basic in how one expects the different isotope measures to reflect atmospheric processes. E.g., is it correct that heavy isotopes are less likely to evaporate from the ocean and hence that atmospheric vapour is lighter than ocean water, i.e., $\delta^2 H$ is negative? Are heavy isotopes more likely to condense on cloud droplets so that condensation in clouds leaves even lighter (more negative $\delta^2 H$) vapor behind? Which physical processes affect the relative variation of $\delta^2 H$ and $\delta^{18}O$? (Is an O-heavy isotope even more unlikely to evaporate than an H-heavy isotope?) In which way are d-excess values or tendencies affected by thermodynamic conditions? How does vertical transport (physically) affect $\delta^2 H$ or d-excess? Maybe it could be helpful add some of the explanation in the caption of Figure 1 to the main text.

*We added a few sentences about what the expected impact of individual moist processes on water isotope signals is:*

- *After introducing water isotopes and the physical basis for their use as tracers for moist diabatic processes in the atmosphere, we added a sentence to explain what the differences in saturation vapour pressure and diffusivity implies at L26: "This implies that the heavy water molecules preferably stay in the condensed phase, where they establish stronger intermolecular bonds compared to their lighter counterpart."*

- *The distinction between equilibrium and non-equilibrium fractionation is introduced explicitly on L28: "Furthermore, the near-surface humidity gradient leads to a differentiation in the relative concentration of the two heavy isotopes ($^1H^2H^{16}O$ and $^1H_2^{18}O$) due to their differences in diffusivity (non-equilibrium fractionation)."*

- Figure 1 was adapted with some visual information on the expected impact of phase change processes on water isotope signals, and the caption was adapted accordingly.

However, we decided to stay relatively short on this aspect and refer the reader to available literature reviews.

2.2 Figure 1: This is a nice illustration. Is it possible to indicate expected dH2 and d-excess values or tendencies in the figure, e.g., by colouring the background according to values or by adding tendencies to the arrows?
We added some visual information about the impact of microphysical processes embedded in the cloud-relative circulation, as recommended by the reviewer. Furthermore, the caption was extended to add some explanations intended for non-isotope experts.

2.3 Figure 1 caption: "can have a depleting or enriching effect" On what does this depend?
There are several factors that need to be taken into account to assess whether below-cloud processes lead to an enrichment or a depletion of the vapour. The partial evaporation of rain droplets will deplete the surrounding vapour due to fractionation (i.e., heavy isotopes preferentially staying in the liquid phase and light isotopes more readily going into the vapour phase). The full evaporation of rain droplets will in most cases enrich the surrounding vapour, since all, heavy and light, isotopes go from the liquid into the vapour phase. However, if the rain droplets formed from much more depleted vapour (i.e., at much higher altitudes) than the surrounding vapour, their full evaporation might also lead to a depletion of the surrounding vapour. Thus, how strong the enrichment/depletion of the surrounding vapour is strongly depends on the original isotopic composition of the rain droplets. These explanations are too long to be added to the caption of Fig. 1. Instead, we now refer to two references (Aemisegger et al., 2015; Graf et al., 2019) where this background about below-cloud processes can be looked up.

2.4 It would be helpful to add a short explanation on why you focus on cloud base properties.
The interest in cloud base comes from the fact that climate models suggest that the cloud base cloud fraction of shallow trade-wind cumuli is very sensitive to changes in environmental conditions, while process models suggest the opposite (Bony et al., 2017, 2022). We have added the following sentence to the introduction on L15: *"Especially, the cloud fraction at cloud base has been identified as a key parameter for the spread of the modelled feedback of these clouds to climate change (Bony et al., 2017)."*

2.5 Section 2.1: Although the simulations are described in Part 1, I would wish for a bit more details here. What are the different domains? Are there limitation due to shallow convection not being resolved at 1 - 10 km grid spacing? Where are trajectories started and which time step do they apply? Can you be more specific in how many trajectories you compute?
**Simulations:** We considerably extended Sect. 2.1 with more information about the simulations (resolution, temporal coverage, nudging) and the convection-resolving model setup.

**Trajectories:**

- Note that details about the trajectories calculated with $COSMO_{iso,5km}$ and $COSMO_{iso,10km}$ are given in the respective section (Sect. 2.4 and 2.5). We now inform about this in Sect. 2.1 on L92:
  *"$COSMO_{iso,1km}$ and $COSMO_{iso,5km}$ are used to characterise the cloud-relative circulation (Sect. 3), and $COSMO_{iso,10km}$ to assess the large-scale circulation (Sect. 4). For this, air parcel backward trajectories are calculated with data from $COSMO_{iso,5km}$ and $COSMO_{iso,10km}$ using the Lagrangian analysis tool LAGRANTO (Wernli and Davies, 1997; Sprenger and Wernli, 2015, see detailed description of trajectory starting points in Sect. 2.4 and Sect. 2.5)."*

- **$COSMO_{iso,5km}$**: We start the backward trajectories at every hourly time step from all *dry-warm* cloud base grid points (red in Fig. 3d) and (as a response to reviewer comment 2.7) at hourly time steps from 1000 randomly selected *clear* cloud base grid points. Thus, the number of *dry-warm* backward trajectories varies from time step to time step. We now show the number in the revised Fig. 2 in terms of fraction (left y-axis) and absolute values (right y-axis). We now also clarify in the caption of Fig. 2 that the *dry-warm* cloud base grid points serve as starting point for backward trajectories. Furthermore, we adapted the text in Sect. 2.4 to the following for clarification (note that the *clear* backward trajectories are new as a response to

reviewer comment 2.7; Fig. 3 was revised accordingly); L153-161:

*"To investigate the formation mechanism of the dry-warm patches at cloud base, we calculate 24 h-backward trajectories. We start them every hourly timestep between 22 January and 13 February 2020 from all cloud base grid points identified as dry-warm in the domain 54.5-61° W and 11-16° N in the $COSMO_{iso,5km}$ simulation. Note that the number of dry-warm cloud base grid points varies from timestep to timestep (Fig. 2). Summed over all timesteps, this results in a total of 568'124 dry-warm trajectories. Similarly, we start 24 h-backward trajectories every hourly timestep between 22 January and 13 February 2020 from 1000 randomly selected cloud base grid points identified as clear in the $COSMO_{iso,5km}$ simulation. We fix the number of clear trajectories for computational reasons, since about 9'000 grid points are identified as clear every hourly timestep (not shown). Summed over all timesteps, this results in a total of 539'460 clear trajectories. The starting points of the dry-warm (red areas) and clear (yellow dots) backward trajectories are shown for an exemplary timestep in Fig. 3d."*

- **$COSMO_{iso,10km}$**: In our opinion, the calculation of the $COSMO_{iso,10km}$ backward trajectories is sufficiently and clearly described in Sect. 2.5. We encourage the reviewer to specify what is unclear, if she disagrees with our opinion.

2.6 L68: "Every vertical profile": do you mean vertical profiles for each grid point?
Yes, each grid point in the considered domain. We changed the sentence to:
"Vertical profiles at every grid point in the domain..."

2.7 L92: Why are dry-warm points of particular interest?
We have added the following text (which also addresses reviewer comment 2.13) to clarify our interest in dry-warm points at the end of Sect. 2.3, L131-151:

*"The reasoning behind the separation of dry-warm and clear cloud base grid points into different categories stems from part of this study. We assume that the characteristics of the dry-warm category result from coherent mesoscale subsidence and, therefore, can give insight into the downward branch of the cloud-relative circulation (sketched in Fig. 1). A dry-warm anomaly is expected at the cloud base level of the downward branch because (1) subsidence causes adiabatic warming (generating a warm anomaly at cloud base), and (2) a coherent mesoscale dry-warm anomaly ensures a certain distance from clouds and through this minimises the influence of mixing with moist air from surrounding clouds thereby avoiding major impacts of evaporating cloud and rain droplets. Crucially, the absence of mixing and phase changes along the subsidence path leads to a conservation of the isotope signal in the vapour from the point at which it was detrained from the cloud down to cloud base. As shown in the exemplary timestep in Fig. 3, our definition of dry-warm indeed applies to grid points that are well away from clouds and therefore are optimal to analyse the processes associated with mesoscale subsidence alone, which we expect to be resolved in the simulations used here.*

*For the clear category, we assume that several processes (subsidence, turbulent mixing, evaporation of cloud and rain droplets) impact its characteristics. Some of these, e.g., turbulent mixing, occur on shorter temporal and spatial scales than resolved by our backward trajectories based on hourly simulation output. Whether the two cloud base environments, clear and dry-warm, truly emerge due to different processes is assessed statistically using backward trajectories as described in the next section.*

*Although we have a special interest in the dry-warm category, it is important to remember that clear cloud base grid points cover a much larger area. Namely, about 81% of the cloud base grid points in $COSMO_{iso,1km}$ are categorised as clear and only about 8% as dry-warm, considered over the whole simulated period. This means, for instance, that for the mass balance at cloud base, the clear category plays a more important role than the dry-warm one (knowing that at the cloud base level, local downward winds of similar magnitude prevail in both; see Villiger et al., 2023, their Fig. 15d)."*

To complement the analysis of dry-warm cloud base environments, we have added some additional analysis of the clear cloud base environments. Namely (as already mentioned in the newly added text above), we have calculated $COSMO_{iso,5km}$ backward trajectories from 1000 randomly selected clear cloud base grid points at every time step. We do so to investigate the different processes leading to the *dry-warm* and *clear* cloud base environments. Furthermore, we also analyse and discuss the conditions in the clear cloud base environments. This has led to the following changes in the manuscript:

- Fig. 3d: Adding of the 1000 randomly selected clear cloud base grid points used to calculate backward trajectories for the illustrative time step shown in the figure. Additionally, we no longer show cloud liquid water in panel (a) and (d), but the grid points identified as *cloud* to clearly show the three different cloud base categories *cloud*, *dry-warm*, and *clear*.
- Sect. 2.4: Adding of the description of the calculation of these 'clear' backward trajectories.
- New Fig. 5 (partially corresponding to former Fig. 5) and associated text: Adding of the analysis of the 'clear' COSMO$_{iso,5km}$ trajectories and discussion of these new results in the text.
- New Fig. 6 (partially corresponding to former Fig. 5) and associated text: Adding of diel cycle of specific humidity and $\delta^2$H in clear cloud base environments and discussion of these new results in the text.

2.8 L94: What is the reasoning to calculate hour-of-the-day deviation from the whole simulation period instead of using day-specific hourly means?

Note that our data has an hourly temporal resolution, which means that the only option to calculate day-specific hourly means is to calculate a mean over space, e.g., over the considered domain (*rel. to domain mean* in Fig. R6 and R7). If we do so, we end up with dry-warm cloud base grid points that reflect the spatial East-West gradient of the cloud base temperature (Fig. R6c), i.e., all dry-warm grid points in the warmer western part of the domain (Fig. R6c).

Here, we are not interested in the East-West gradient of cloud base temperature, but in the mesoscale subsidence (leading to the dry-warm cloud base grid points; see also answer to reviewer comment 2.7) occurring around clouds everywhere in the considered domain. Therefore, we decided to define the dry-warm characteristics relative to grid-points wise mean values. We can define grid-point wise mean values (1) over the whole period (e.g., one mean value per grid point; *rel. to overall mean* in Fig. R6 and R7) or (2) for each hour-of-the-day over the whole period (e.g., 24 mean values per grid point; *rel. to hour-of-day mean* in Fig. R6 and R7).

Using the definition *rel. to overall mean* yields slightly more *dry-warm* grid points (8.6 % over the whole period) than using the definition *rel. to hour-of-day mean* (8 % over the whole period). Thus, the *rel. to hour-of-day mean* definition might be seen as the more restrictive one. The spatial (Fig. R6a,b) as well as the temporal (Fig. R7) distribution of the two differently identified *dry-warm* cloud base grid points is very similar, leading to the conclusion that the distinction between the two definitions is not decisive for the key findings of our study. Based on this and for consistency with part 1 of this study (Villiger et al., 2023), we have not adapted the definition applied in the paper.

2.9 L105: How do you choose starting points? Or do you start from all dry-warm cloud base grid points?

See answer to reviewer comment 2.5.

2.10 The naming convention of variables in Fig. 4 is not clear to me.

We adapted the naming convention of the variables in Fig. 4 and reformulated the figure caption for more clarity. Note that the figure was changed additionally due to our response to reviewer 1, major comment B.

2.11 L126: How do vertical velocities between the 1-km and the 5-km compare? Do cloudy updraft strengths differ?

Cloudy updraft (i.e., named *mass flux* in Fig. R8) strengths are stronger in COSMO$_{iso,1km}$ than in COSMO$_{iso,5km}$ (Fig. R8a). However, most importantly, the diel cycles are similar in the two datasets (Fig. R8b). The same is true for the negative velocities in *dry-warm* cloud base grid points (named *negative mass flux* in Fig. R8; which might be seen as the downward branch of the cloud-relative overturning circulation, see also reviewer comment 2.7): Downward velocities in *dry-warm* cloud base grid points are stronger in the COSMO$_{iso,1km}$ data than in the COSMO$_{iso,5km}$ data (Fig. R8c), but the diel cycles are very similar (Fig. R8d). We do not think that the differences of updraft strengths between the COSMO$_{iso,5km}$ and COSMO$_{iso,1km}$ are relevant for our findings, because we are not primarily interested in their magnitude but rather in their variation over the course of the day and the impact of this variation on cloud base isotopes (i.e., the coupling between the variables). We now clarify this at the beginning of Sect. 3., L200-204:

*"When combining these two datasets, it is important to know that the vertical velocities within cloud and dry-warm cloud base environments are stronger in COSMO$_{iso,1km}$ than in COSMO$_{iso,5km}$.*

[Figure]

**Figure R6:** Spatial distribution of **(a-c)** cloud base grid points identified as *dry-warm*, **(d)** cloud liquid water (QC), **(e)** specific humidity (QV), and **(f)** potential temperature ($\theta$) at cloud base at 15 UTC on 2 February 2020. *Dry-warm* cloud base grid points are identified as deviations in specific humidity and potential temperature (see paper for details) relative to **(a)** the domain mean of the current time step, **(b)** the grid-point-wise mean over the whole period, or **(c)** the grid-points-wise hour-of-day mean over the whole period. The fraction of grid points in the domain identified as *dry-warm* is **(a)** 4.3 %, **(b)** 6.3 %, and **(c)** 9.8 %. Shown is the data from COSMO$_{iso,5km}$ in the domain 54.5-61° W and 11-16° N and the track flown by the research aircraft ATR (cf. Villiger et al., 2023) in pink for scale. Note that similar results are obtained with the data from COSMO$_{iso,1km}$. Data: COSMO$_{iso,5km}$.

[Figure]

**Figure R7:** Hourly time series of the fraction (left axis) and number (right axis) of cloud base grid points identified as *dry-warm* in the domain 54.5-61° W and 11-16° N for COSMO$_{iso,5km}$. Three different definitions of *dry-warm* are applied (see response to reviewer comment 2.8). Data: COSMO$_{iso,5km}$.

*However, the diel cycles of vertical velocities in the two datasets are the same (not shown). We argue that we can use the variables characterising the cloud-relative circulation derived from the COSMO$_{iso,5km}$ trajectories, as our primary focus lies in discerning their variation (and not their absolute values) throughout the day and in understanding how this variation correlates with changes in cloud base isotopes."*

[Figure]

**Figure R8: (a,b)** Mass flux (mean vertical velocity of all cloudy cloud base grid points with positive vertical velocities) **(c,d)** "negative mass flux" (mean vertical velocity of all dry-warm cloud base grid points with negative vertical velocities; only time steps with at least 100 grid points are considered) over whole simulated period for COSMO$_{iso,1km}$ (red) and COSMO$_{iso,5km}$ (red). Shown are **(a,c)** hourly values and **(b,d)** the diel cycle (25-75 percentile range as shading and the median as line). Calculated with data from domain 54.5-61° W and 11-16° N during the period 20 January to 13 February 2020 are shown.

2.12 L137: "largely unaffected": Please explain how you arrive at this interpretation. Because the signal is small?
Yes, exactly. The BCO mixed layer $q$ and $\delta^2H$ are in phase and of about the same amplitude as the cloudy cloud base signals. We clarified this in the text with the sentence stated in the response to reviewer comment 1.4.

2.13 L140: Can you check how well it is closed by multiplying area fractions and vertical velocities of both branches?

From part 1 of this study (Table 4 and Fig. 15), we know the fraction of cloud base grid points categorised as *dry-warm*, *clear*, (non-precipitating) *cloud*, and (precipitating) *cloud-rain* as well as the mean vertical velocity for each of these categories. We can use these numbers for a first rough mass balance analysis (Table R2). The mass fluxes at cloud base are not fully balanced in $COSMO_{iso,1km}$ considered over the domain 54.5-61° W and 11-16° N and over the period 20 January to 13 February 2020. Adding the estimated mass fluxes from the different cloud base categories (Table R2), we get a negative term of $-387'637 \times 10^6$ m$^3$ s$^{-1}$. Note that this is a very rough estimate, since we summarized mass fluxes over a considerably large domain and time period. However, the key message of the analysis is that the upward mass flux of clouds (*cloud* and *cloud-rain*) is fully balanced by the downward mass flux occurring in the *clear* cloud base grid points. The contribution of the *dry-warm* cloud base grid points to the negative mass flux is negligible in comparison. The dominance of the *clear* category over the *dry-warm* category in terms of negative mass flux comes from its much larger areal coverage.

Considering the importance of the *clear* cloud base category for the mass balance, we add it to our analysis of the diel cycle in Sect. 3. Please also refer to our response to reviewer comment 2.7.

**Table R2:** Fraction and number of cloud base grid points identified as *dry-warm*, *clear*, *cloud*, and *cloud-rain* in the $COSMO_{iso,1km}$ simulation in the domain 54.5-61° W and 11-16° N during the period 20 January to 13 February 2020. For each cloud base category, the mean vertical velocity is calculated and the mass flux (as the area corresponding to the number of 1×1 km grid points multiplied by the mean vertical velocity. Note that we distinguish between precipitating clouds (*cloud-rain*) and non-precipitating clouds (*cloud*) because the analysis shown here is taken from part 1 of this study (cf. Table 4 and Fig. 15 in Villiger et al., 2023).

| Cloud base category | Fraction of grid points [%] | Number of grid points | Mean vertical velocity [m s$^{-1}$] | Mass flux [×10$^6$ m$^3$ s$^{-1}$] |
|---|---|---|---|---|
| *dry-warm* | 7.6 | 14'262'075 | -0.01 | $-142'620.75$ |
| *clear* | 81.2 | 152'943'787 | -0.02 | $-3'058'875.74$ |
| *cloud* | 10.9 | 20'597'884 | 0.12 | $2'471'746.08$ |
| *cloud-rain* | 0.3 | 645'497 | 0.53 | $342'113.41$ |

2.14 L144: It is not clear to me why "the amount of vapour that returns to the cloud base is less"? Because it is colder higher up and therefore saturated q is less?

Yes. We clarified this with the adapted text on L274-281:

*"(2) A dominating temporally delayed effect due to the vertical growth of clouds, which has the consequence that the detrainment from clouds happens at increasing altitudes where temperature is lower and, therefore, saturated q is less. With increasing altitudes, the isotope signal is also more depleted due to the continuous condensation and rainout in the convective updrafts which transport the vapour to these altitudes in the first place. Assuming that the vapour detrained from clouds experiences little to no phase changes or mixing with advected vapour during its journey back to cloud base (i.e., closed overturning circulation), it follows that its isotope signal is approximatively conserved. Thus, the higher the detrainment from clouds, the lower the amount of vapour that returns to cloud base, and the more depleted its isotope signal."*

2.15 149-153: This is not clear to me. Why does the min-max time shift in dry-warm patch characteristics determine the point in time when subsiding air detrained from the cloud?

Please refer to our response to reviewer 2 major point B. With our new approach of looking for each trajectory's cloud detrainment altitude instead of the subsidence rate over 12 h, we hope to have clarified this point of confusion.

2.16 L154-159: How do your results compare to literature, e.g. observed circulations from George et al. or the diurnal cycle from Vial et al.?

We implemented several comparisons to Vial et al. (2019), Vogel et al. (2020,2022), and George et al. (2023) in Sect. 3.

2.17 Fig. 6 and 8 are barely discussed in the text.

Fig. 6 (now Fig. 7) is discussed in Sect. 4 on L315-316, L339, L345. Figure 8 (now Fig. 10) is now additionally discussed in Sect. 5, L378-381:

*"In Fig. 10, we summarize the main findings of this study. Namely, the values of q and δ²H in dry-warm cloud base environments are determined (a) by the cloud-relative circulation through the*

*cloud detrainment altitude (reflecting the diel cycle of convection) as well as (b) by the large-scale subsidence reflecting different circulation patterns with distinct moisture source conditions."*
Furthermore, Fig. 10 is also discussed on L387-389 (see response to reviewer comment 1.11).

2.18 L200: Why does a deeper circulation need to be stronger?
Because we looked at it over a constant period of time (i.e., 12h). A stronger subsidence over 12 h implies that the air parcels originate from higher altitude. This sentence was removed from the manuscript due to the new approach with the cloud detrainment altitude (see also response to comment 1.5).

2.19 L204: Do you refer to the few points with low d-excess around q = 8-9 g/kg here? They don't seem special to me in terms of large-scale subsidence (Fig. 8b).
No, we refer to the points with strong large-scale subsidence (most of them from 22 January 2020; encircled black dashed in Fig. R9). We now clarify this by encircling them in the figure and refer to them in the text correspondingly.

[Figure]

**Figure R9:** Same as Fig. 8b of the paper, but with the data points of interest highlighted by encircling them (black dashed).

**Technical comments**:
Please check figures, their captions and references in the text carefully.
We checked this thoroughly.

I noted a few typos/corrections below as technical comments.

2.1 L76: "761, 914, and 1082m" this is not clear
We changed this sentence to *"The hourly cloud base levels alternate between 783 and 970 m for $COSMO_{iso,10km}$ and change between three levels, i.e. 761, 914, and 1082 m, for $COSMO_{iso,5km}$ and $COSMO_{iso,1km}$."*

2.2 L107: There is no Fig 4b Done.

2.3 Fig. 2: caption and axis labels do not match This conflict has been resolved.

2.4 Fig. 3: add "(red)" after "dry-warm"; (b) → (d) Done, both as suggested.

2.5 Fig. 5: red and green as contrasting colours are not optimal for colour-blind readers; in c please use dashed line style also to distinguish the thin lines
In Fig. 6 we changed the Figure as suggested and used blue instead of green for the clouds. The isotope signals are also clearly stratified along the vertical axis such that a confusion of the groups can be avoided.

2.6 Fig. 7: What do the two different dates per plot mean?
   The black date on the left of the panels show the arriving times, the grey dates 3.5 days before arrival. We clarified this in the caption.

2.7 L135: delete "d-e" Done.

**References**

[revised manuscript text omitted]